# Continuum of care for maternal health in Uganda: A national cross-sectional study

Quraish Sserwanja[1], David Mukunya[2,3], Prossy Nabachenje[4], Alleluyah Kemigisa[3], Paul Kiondo[5], Julius N. Wandabwa[6], Milton W. Musaba[6]*

1 Programs Department, GOAL, Khartoum, Sudan, 2 Department of Community and Public Health, Busitema University, Mbale, Uganda, 3 Sanyu Africa Research Institute, Mbale, Uganda, 4 Department of Paediatrics and Child Health, Busitema University, Mbale, Uganda, 5 Department of Obstetrics and Gynaecology, Makerere University, Kampala, Uganda, 6 Department of Obstetrics and Gynaecology, Busitema University, Mbale, Uganda

* miltonmusaba@gmail.com

## Abstract

### Introduction

A continuum of maternal care approach can reduce gaps and missed opportunities experienced by women and newborns. We determined the level of coverage and factors associated with the continuum of maternal care in Uganda.

### Methods

We used weighted data from the Uganda Demographic and Health Survey (UDHS) 2016. We included 10,152 women aged 15 to 49 years, who had had a live birth within five years preceding the survey. Stratified two-stage cluster sampling design was used to select participants. Continuum of maternal care was considered when a woman had at least four antenatal care (ANC) visits, had delivered in a health facility and they had at least one postnatal check-up within six weeks. We conducted multivariable logistic regression analysis to determine factors associated with completion of the continuum of maternal care using SPSS version 25.

### Results

The level of coverage of complete continuum of maternal care was 10.7% (1,091) (95% CI: 10.0–11.2). About 59.9% (6,080) (95% CI: 59.0–60.8) had four or more antenatal visits while 76.6% (7,780) (95% CI: 75.8–77.5) delivered in a health facility and 22.5% (2,280) (95% CI: 21.5–23.2) attended at least one postnatal care visit within six weeks. The following factors were associated with continuum of maternal care; initiating ANC in the first trimester (AOR 1.49, 95% CI: 1.23–1.79), having secondary level of education (AOR 1.60, 95% CI: 1.15–2.22) and tertiary level of education (AOR 2.08 95% CI: 1.38–3.13) compared to no formal education, being resident in Central Uganda (AOR 1.44, 95% CI:1.11–1.89), Northern Uganda (AOR 1.35, 95% CI: 1.06–1.71) and Western Uganda (AOR 0.61, 95% CI: 0.45–0.82) compared to Eastern Uganda, and exposure to newspapers and magazines.

**Data Availability Statement:** All relevant data are within the paper and its Supporting Information files.

**Funding:** The author(s) received no specific funding for this work.

**Competing interests:** The authors have declared that no competing interests exist.

## Conclusion

The level of coverage of the complete continuum of maternal care was low and varied across regions. It was associated with easily modifiable factors such as early initiation of ANC, exposure to mass media and level of education. Interventions to improve utilisation of the continuum of maternal care should leverage mass media to promote services, especially among the least educated and the residents of Western Uganda.

## Introduction

Globally, huge disparities exist in the distribution of maternal morbidity and mortality [1, 2]. Currently, most global maternal deaths occur in sub-Saharan Africa [2, 3]. At 1 in 59 pregnancies, sub-Saharan Africa has the highest lifetime risk of maternal death, compared to 1 in 160 pregnancies for the rest of the low-income countries [4] and one (1) in 5,400 pregnancies for high income countries [5]. Furthermore, at 27 deaths per 1,000 live births, sub-Saharan Africa has the highest neonatal mortality in the world [6]. Majority of the maternal and perinatal morbidity and mortality occurs in the intrapartum and immediate postpartum period [2, 3, 7]. Uganda's perinatal mortality has stagnated at 40 per 1,000 pregnancies over the last decade [8, 9]. Over the same period, the utilization of maternal healthcare services has generally improved, but maternal mortality rate has remained unacceptably high at 336 maternal deaths per 100,000 live births [9, 10]. It is estimated that 80% of maternal deaths and 66% of neonatal deaths in the world could be averted by ensuring that women have access to quality and effective maternal healthcare services especially during childbirth and in the first week of life [11, 12]. Therefore, strengthening and improving access to the complete continuum of maternal care presents a unique opportunity to prevent most of the direct causes of death [12–15].

The continuum of maternal healthcare approach emphasizes access to comprehensive, integrated and continuous interventions throughout the pregnancy, childbirth, postnatal period, infancy, childhood and the pre-pregnancy period [13, 16]. Exposure to the entire spectrum of maternal care ensures that utilisation at each stage builds on the effect of the previous stage of care [13]. This strategy can effectively reduce the gaps and missed opportunities experienced by women with the resultant potential to save the most lives of mothers, newborns and children [17]. In conformity with the current World Health Organisation (WHO) guidelines, since January 2021 [18], the Ministry of Health in Uganda recommends that all women attend at least eight antenatal care (ANC) visits and have childbirth in a health facility supervised by a skilled service provider [19]. The same guidelines recommend that a skilled health worker should examine all women and their newborns as early as possible for timely identification and treatment of any complications that might limit survival [17]. For home births, the first postnatal check should occur within 24 hours of birth. Thereafter, all mothers and newborns should receive at least three additional postnatal checks on day three, between day seven and fourteen, and between week two and six following child birth [20, 21]. In Uganda, the factors associated with the completion of continuum of maternal care have not been well documented. To accelerate progress towards achieving the set targets for maternal child health in the sustainable development goal three (SDG #3) [22] and the every new born action plan (ENAP) [23], it is vital to identify ways of addressing this mismatch between the significant improvement in the utilisation of maternal healthcare services and the unsatisfactory maternal and child health (MCH) indicators.

Factors associated with each of the individual MCH healthcare services such as ANC, skilled birth attendance (SBA) and postnatal care (PNC) are well known [24–26]. However, nationally representative data on the level of coverage and determinants of exposure to the

complete continuum of maternal care is limited. The current study focused on the completion of continuum of maternal healthcare by specifically looking at ANC attendance, health facility utilization during childbirth and PNC. We aimed to estimate the level of coverage and factors associated with completion of the continuum of maternal healthcare in Uganda. This information might inform the design of interventions and strategies to promote utilisation of the complete continuum of maternal care, and thereby improve the MCH indicators.

## Materials and methods

### Study setting

The Uganda Demographic and Health Survey (UDHS) collected data on women's sociodemographic characteristics, reproductive health and nutrition indicators [27, 28]. The UDHS was implemented by the Uganda Bureau of Statistics with the technical assistance of Inner City Fund International through the USAID-supported MEASURE DHS project [29].

Uganda has one of the youngest populations in world, with about 85% below the age of 15 years. Approximately, 22% (7.3 million) of the population are women of reproductive age 15–49 years, with a high total fertility rate of 5.1 births per woman. Virtually all pregnant mothers access ANC services at least once and 60% at least four times [18, 29]. Currently, institutional deliveries are at 74%. Family planning service utilisation has also improved albeit too slowly with the current contraceptive prevalence rate of 39% [29]. In Uganda, access to emergency obstetric and neonatal care services provided by a skilled birth attendant has improved significantly [9]. According to the latest UDHS survey, most women sought care from a nurse or midwife (84%), 9% received care from a doctor, and only 1% received ANC services from a traditional birth attendant [9].

In Uganda, health services are provided by the public and private sub-sectors, with each sub-sector contributing about 50 percent of the service delivery outlets [30]. The public sector includes national and regional hospitals and a tiered system of health centres which handle a range of services from community outreach (Health Centre [HC] 1's or Voluntary Health Committees) to cesarean sections (HC IVs). Each health sub-district with an estimated population of 100,000 people is served by a HC IV facility, above this is a general district hospital and then a regional referral hospital with specialized services. The private health service providers comprise private not-for-profit organizations, private for-profit health care providers also known as commercial health care providers, and traditional and complementary medicine practitioners. Tranditional medicine practitioners significantly contribute to the provision of health care services in Uganda [31]. A recent WHO report estimates that the ratio of traditional medicine practitioners to population is between 1:200 and 1:400 compared with a doctor to population ratio of 1:18,000, which implies a significant contribution to the human resources for health services [32]. In the health sector, the decentralization policy has greatly improved physical access because it is estimated that 72 percent of the population lives within 5 km of a health facility.

### Study design

This study is the secondary data analysis of the nationally-representative cross-sectional study (UDHS) conducted between June and December 2016.

### Study population

Our secondary data analysis used information collected using validated women's and household questionnaires [29]. We included women aged 15–49 years who had a live birth within

five years preceding the survey and had given informed consent. The UDHS interviewed a total of 18,506 women aged 15–49 years [29].

## Sample size and sampling

Our secondary data analysis was a total enumeration of all the 10,152 women who had had a live birth within five years preceding the survey [29].

## Eligibility criteria

We included women aged 15 to 49 years who either were permanent residents or slept in the selected household the night before the survey and had given a live birth in the previous five years.

## Study variables

**Outcome variable.** Our outcome variable was a complete continuum of maternal care. Its definition was based on the WHO and Ministry of Health recommendations at the time of conducting this survey. It was considered when a woman reported having had: at least four ANC (ANC4+) [18] contacts during the most recent pregnancy, utilized a health facility during the most recent childbirth and their newborn had at least one postnatal check-up within six weeks after discharge from place of childbirth [11, 33, 34]. These were constructed into binary variable complete coded as one (1) and incomplete coded as zero (0). Similar analysis of the outcome variable was also done to look at the level of completion of the continuum of maternal care using the 2001, 2006 and 2011 UDHS. The details are in S1 Table.

**Independent variables.** Independent variables were categorized into women, partners' and household characteristics which were chosen basing on previous studies [11, 14, 24, 34–38] and availability in the UDHS data base. The women's characteristics were age (15–24, 25–34, and 35–49) in completed years, level of education (no education, primary, secondary, and tertiary), exposure to media (Newspapers, magazines, radio and television (at least once a week, less than once a week and not at all). The others were parity (para1, 2–4 and 5+), age at birth (less than 18 and 18 and above), ANC timing (first trimester and above first trimester), marital status (married and not married), working status (working and not working), decision making for seeking healthcare (woman alone, woman with someone and not involved), and religion (Muslims, Catholics, Anglican, Pentecostal, and others).The partner's characteristics included the highest level of education (no education, primary, secondary, and tertiary) attained. The household characteristics included the wealth index of household (categorized into quintiles: richest, richer, middle poorer and poorest), type of residence (urban and rural), sex of household head (male and female), size of household (less than 6 and 6 and above) and region that included North, East, West and Central.

## Statistical analysis

We performed complex sample analysis using SPSS version 25.0 statistical software to account for the multi-stage cluster study design and weighted data was used to account for the unequal probability sampling in different strata. We tabulated proportions and frequencies for all the independent variables. We conducted a bivariable logistic regression analysis to assess the association of each independent variable with complete continuum of maternal care. We presented crude odds ratio (COR), 95% confidence interval (CI) and p-values. Independent variables found significant at p-value less than 0.25 [39] in the bivariable analysis and those known to be associated from the literature as summarized in the conceptual framework by Owili et al.

[40] were included in the multivariable models. We prepared two adjusted models with model one including variables that were significant at bivariable level and had no missing data while second model included all variables in model one and those that had missing data (husband's level of education). Adjusted odds ratios (AOR), 95% CI and p-values were calculated with statistical significance level set at p-value < 0.05. Results of the first model which is the main model are shown in Table 3 and results of the second model are presented in S2 Table.

## Results

### Characteristics of the study population

The mean age of women was 28.5 [standard deviation (SD) 7.1] years with majority (43.6%) of them being 25–34 years. Most of the women were residents in rural areas (76.9%), were married (81.3%), working (79.0%), and initiated ANC after the first trimester (70.9%). Of the 10,152 women, 59.9% (6,080) had four or more ANC contacts while 76.6% (7,780) utilized health facility for childbirth and 22.5% (2,280) attended newborn PNC within six weeks after childbirth. Overall, 10.7% (1,091) had a complete continuum of maternal care. The details are shown in Table 1. Among women who delivered at home, 29.4% had newborn PNC compared to 20.3% among those who delivered at a health facility. The details are shown in Table 2. Between the 2011 and 2016 UDHS, the level of the complete continuum maternal care declined by only one percentage point. Details are in S1 Table.

### Factors associated with complete continuum of maternal care

Results from multivariable logistic regression analysis (adjusted model 1, Table 3) showed that region, timing of ANC initiation, exposure to newspapers, timing of ANC initiation, and women's level of education were associated with complete continuum of maternal care. Women from Central Uganda (AOR 1.44, 95% CI:1.11–1.89) and those from Northern Uganda (AOR 1.35, 95% CI: 1.06–1.71) were more likely to have complete continuum of maternal care compared to their counterparts in Eastern Uganda. Women from Western Uganda (AOR 0.61, 95% CI: 0.45–0.82) were less likely to have complete continuum of maternal care. Women with secondary level of education (AOR 1.60, 95% CI:1.15–2.22) and those with tertiary level of education (AOR 2.08 95% CI: 1.38–3.13) were more likely to have complete continuum of maternal care compared to their counterparts with no education. Women initiating ANC in the first trimester (AOR 1.49, 95% CI:1.23–1.79) were more likely to have complete continuum of maternal care compared to their counterparts who initiated ANC after the first trimester. Exposure to newspapers less than once a week (AOR 1.33, 95% CI:1.05–1.69) and at least once a week (AOR 1.97, 95% CI:1.53–2.56) was positively associated with complete continuum of maternal care.

## Discussion

This study used the 2016 UDHS data to investigate the level and determinants of access to the complete continuum of maternal care. We found a low level of utilization of complete continuum of maternal care. Initiation of ANC in the first trimester, being residents in the central and western regions, level of education and exposure to mass media were associated with completion of the continuum of maternal care.

In Uganda, about one in 10 women completed the continuum of maternal care. This is similar to that of other countries in the region, and has not changed much since the previous survey done in 2011 (S1 Table). Similar studies based on secondary data from nationally representative samples have reported a 9.1% level of coverage in Ethiopia [35], and 38% level

**Table 1. Background characteristics of women who had given birth in the last 5 years as per the 2016 UDHS.**

| Category | Frequency (N = 10,152) | Percent (%) |
|---|---|---|
| **Age** | | |
| 15 to 24 | 3546 | 34.9 |
| 25 to 34 | 4425 | 43.6 |
| 35 to 49 | 2181 | 21.5 |
| **Residence** | | |
| Urban | 2346 | 23.1 |
| Rural | 7806 | 76.9 |
| **Region** | | |
| Western | 2559 | 25.2 |
| Eastern | 2727 | 26.9 |
| Central | 2805 | 27.6 |
| Northern | 2061 | 20.3 |
| **Parity** | | |
| 5 and above | 3453 | 34.0 |
| 2–4 | 4650 | 45.8 |
| 1 | 2049 | 20.2 |
| **Household Size** | | |
| 6 and above | 5062 | 49.9 |
| Less than 6 | 5090 | 50.1 |
| **Working status** | | |
| Not working | 2136 | 21.0 |
| Working | 8016 | 79.0 |
| **Marital status** | | |
| Married | 8256 | 81.3 |
| Not married | 1896 | 18.7 |
| **Education Level** | | |
| No Education | 1061 | 10.5 |
| Primary Education | 6091 | 60.0 |
| Secondary Education | 2285 | 22.5 |
| Higher | 715 | 07.0 |
| **Wealth Index** | | |
| Poorest | 2117 | 20.9 |
| Poorer | 2074 | 20.4 |
| Middle | 1921 | 18.9 |
| Richer | 1862 | 18.3 |
| Richest | 2178 | 21.5 |
| **Exposure to Radio** | | |
| Not at all | 2668 | 26.3 |
| Less than once a week | 1551 | 15.3 |
| At least once a week | 5933 | 58.4 |
| **Exposure to Newspapers** | | |
| Not at all | 8188 | 80.6 |
| Less than once a week | 1209 | 11.9 |
| At least once a week | 755 | 7.5 |
| **Exposure to television** | | |
| Not at all | 7210 | 71.0 |
| Less than once a week | 1105 | 10.9 |

(*Continued*)

**Table 1.** (Continued)

| Category | Frequency (N = 10,152) | Percent (%) |
|---|---|---|
| At least once a week | 1837 | 18.1 |
| **Husband's education level[a]** | | |
| None | 517 | 5.1 |
| Primary | 4346 | 42.8 |
| Secondary | 2205 | 21.7 |
| Tertiary | 975 | 9.6 |
| **Sex of household head** | | |
| Female | 2726 | 26.9 |
| Male | 7426 | 73.1 |
| **Timing of ANC** | | |
| 1–3 months | 2953 | 29.1 |
| Above 3 months | 7199 | 70.9 |
| **Age at first birth** | | |
| Less than 18 | 3770 | 37.1 |
| 18 and above | 6382 | 62.9 |
| **Religion** | | |
| Muslim | 1409 | 13.9 |
| Others | 301 | 3.0 |
| Anglican | 3154 | 31.1 |
| Pentecostal | 1285 | 12.7 |
| Catholics | 4003 | 39.4 |
| **Healthcare seeking decision[b]** | | |
| Not involved | 2261 | 22.3 |
| Woman and Someone | 3623 | 35.7 |
| Woman alone | 2372 | 23.4 |

[a]Missing 2109
[b]Missing 1896.

**Table 2. Level of utilization of the various maternal and child health services.**

| Category | Frequency | Percentage (95% CI) |
|---|---|---|
| **ANC contacts** | | |
| 4 and above | 6080 | 59.9 (59.0–60.8) |
| Less than 4 | 4072 | 40.1 (39.2–41.0) |
| **Place of childbirth** | | |
| Health facility | 7780 | 76.6 (75.8–77.5) |
| Home | 2372 | 23.4 (22.5–24.2) |
| **Postnatal care** | | |
| Yes | 2280 | 22.5 (21.5–23.2) |
| No | 7872 | 77.5 (76.8–78.4) |
| **Complete continuum of maternal care** | | |
| Yes | 1091 | 10.7 (10.0–11.2) |
| No | 9061 | 89.3 (88.8–90.0) |

**Table 3. Predictors of complete continuum of maternal care in Uganda.**

| Category | Incomplete COC n = 9061 | Complete COC n = 1091 | Crude model COR (95% CI) | P-value | Adjusted model I AOR (95%CI) N = 10,152 |
|---|---|---|---|---|---|
| | n (%) | n (%) | | | |
| **Age** | | | | 0.002 | |
| 35 to 49 | 2004 (22.1) | 178 (16.3) | 1 | | 1 |
| 25 to 34 | 3897 (43.0) | 528 (48.4) | **1.53 (1.20–1.95)** | | 1.17 (0.90–1.51) |
| 15 to 24 | 3160 (34.9) | 385 (35.3) | **1.38 (1.11–1.71)** | | 1.00 (0.73–1.36) |
| **Residence** | | | | <**0.001** | |
| Urban | 1968 (21.7) | 378 (34.6) | 1 | | 1 |
| Rural | 7093 (78.3) | 713 (65.4) | **0.52 (0.42–0.66)** | | 1.04 (0.82–1.33) |
| **Region** | | | | <0.001 | |
| East | 2485 (27.4) | 242 (22.2) | 1 | | 1 |
| North | 1846 (20.4) | 215 (19.7) | 1.20 (0.95–1.50) | | **1.35 (1.06–1.71)** |
| West | 2414 (26.6) | 145 (13.3) | 0.62 (0.46–0.83) | | **0.61 (0.45–0.82)** |
| Central | 2316 (25.6) | 490 (44.9) | **2.17 (1.68–2.82)** | | **1.44 (1.11–1.89)** |
| **Parity** | | | | <0.001 | |
| 5 and above | 3190 (35.2) | 263 (24.1) | 1 | | 1 |
| 2–4 | 4111 (45.4) | 539 (49.4) | **1.59 (1.32–1.92)** | | 1.10 (0.87–1.38) |
| 1 | 1760 (19.4) | 289 (26.5) | **1.99 (1.58–2.50)** | | 1.33 (0.96–1.84) |
| **Household size** | | | | <0.001 | |
| 6 and above | 4588 (50.6) | 474 (43.4) | 1 | | 1 |
| Less than 6 | 4473 (49.4) | 617 (56.6) | **1.34 (1.16–1.55)** | | 1.10 (0.94–1.28) |
| **Working status** | | | | 0.615 | |
| Working | 7145 (78.9) | 871 (79.8) | 1 | | - |
| Not working | 1916 (21.1) | 220 (20.2) | 0.94 (0.76–1.18) | | - |
| **Marital status** | | | | 0.680 | |
| Not Married | 1698 (18.7) | 198 (18.1) | 1 | | - |
| Married | 7363 (81.3) | 894 (81.9) | 1.04 (0.86–1.27) | | - |
| **Education level** | | | | <0.001 | |
| No education | 994 (11.0) | 68 (6.2) | 1 | | 1 |
| Primary Education | 5606 (61.9) | 484 (44.4) | **1.26 (0.96–1.65)** | | 1.15 (0.87–1.51) |
| Secondary Education | 1934 (21.3) | 351 (32.1) | **2.64 (1.99–3.51)** | | **1.60 (1.15–2.22)** |
| Tertiary | 527 (5.8) | 188 (17.2) | **5.19 (3.69–7.29)** | | **2.08 (1.38–3.13)** |
| **Wealth Index** | | | | <0.001 | |
| Richest | 1753 (19.3) | 425 (39.0) | 1 | | 1 |
| Richer | 1658 (18.3) | 204 (18.7) | **0.51 (0.39–0.66)** | | 0.92 (0.68–1.24) |
| Middle | 1785 (19.7) | 136 (12.5) | **0.31 (0.24–0.41)** | | 0.74 (0.55–1.01) |
| Poorer | 1921 (21.2) | 153 (14.0) | **0.33 (0.26–0.42)** | | 0.79 (0.58–1.08) |
| Poorest | 1944 (21.5) | 173 (15.9) | **0.37 (0.29–0.47)** | | 0.86 (0.61–1.21) |
| **Exposure to radio** | | | | <**0.001** | |
| **Not at all** | **2458 (27.1)** | **209 (19.2)** | **1** | | **1** |
| **Less than once a week** | **1384 (15.3)** | **167 (15.3)** | **1.42 (1.10–1.84)** | | **1.11 (0.85–1.44)** |
| **At least once a week** | **5219 (57.6)** | **715 (65.5)** | **1.61 (1.34–1.94)** | | **1.11 (0.91–1.36)** |
| **Exposure to newspapers** | | | | <0.001 | |
| Not at all | 7501 (82.8) | 686 (62.9) | 1 | | 1 |
| Less than once a week | 1008 (11.1) | 201 (18.4) | **2.18 (1.73–2.75)** | | **1.33 (1.05–1.69)** |
| At least once a week | 552 (6.1) | 204 (18.7) | **4.04 (3.32–4.92)** | | **1.97 (1.53–2.56)** |
| **Exposure to television** | | | | <0.001 | |

*(Continued)*

**Table 3.** (Continued)

| Category | Incomplete COC n = 9061 | Complete COC n = 1091 | Crude model COR (95% CI) | P-value | Adjusted model I AOR (95%CI) N = 10,152 |
|---|---|---|---|---|---|
| | n (%) | n (%) | | | |
| Not at all | 6628 (73.1) | 582 (53.3) | 1 | | 1 |
| Less than once a week | 965 (10.7) | 140 (12.8) | **1.65 (1.31–2.08)** | | 1.11 (0.86–1.44) |
| At least once a week | 1468 (16.2) | 369 (33.8) | **2.86 (2.31–3.56)** | | 1.29 (0.98–1.70) |
| **Husband's education level**[a] | | | | <0.001 | |
| None | 474 (6.6) | 43 (5.0) | 1 | | - |
| Primary | 4015 (55.9) | 331 (38.4) | 0.92 (0.65–1.31) | | - |
| Secondary | 1922 (26.8) | 283 (32.8) | **1.64 (1.13–2.39)** | | - |
| Tertiary | 769 (10.7) | 206 (23.9) | **2.98 (2.03–4.36)** | | - |
| **Sex of household head** | | | | 0.444 | |
| Male | 6642 (73.3) | 783 (71.8) | 1 | | - |
| Female | 2419 (26.7) | 308 (28.2) | 1.08 (0.89–1.31) | | - |
| **Timing of ANC** | | | | <0.001 | |
| Above 3 months | 6517 (71.9) | 396 (36.4) | 1 | | 1 |
| 1–3 months | 2544 (28.1) | 695 (63.6) | **1.46 (1.22–1.76)** | | **1.49 (1.23–1.79)** |
| **Age at first birth** | | | | 0.001 | |
| 18 and above | 5635 (62.2) | 747 (68.5) | 1 | | 1 |
| Less than 18 | 3426 (37.8) | 344 (31.5) | **0.76 (0.64–0.90)** | | 1.04 (0.87–1.24) |
| **Religion** | | | | 0.118 | |
| Catholics | 3576 (39.5) | 427 (39.1) | 1 | | 1 |
| Others | 270 (3.0) | 32 (2.9) | 0.98 (0.64–1.49) | | 1.11 (0.72–1.70) |
| Anglican | 2832 (31.3) | 322 (29.5) | 0.95 (0.80–1.14) | | 0.99 (0.83–1.20) |
| Pentecostal | 1163 (12.8) | 122 (11.2) | 0.88 (0.68–1.14) | | 0.78 (0.59–1.03) |
| Muslims | 1220 (13.5) | 189 (17.3) | **1.30 (1.03–1.64)** | | 1.04 (0.819–1.34) |
| **Care seeking decision** | | | | 0.476 | |
| Woman alone | 2099 (28.5) | 273 (30.6) | 1 | | - |
| Woman and Someone | 3234 (43.9) | 389 (43.6) | 0.93 (0.77–1.11) | | - |
| Not involved | 2030 (27.6) | 231 (25.9) | 0.88 (0.70–1.10) | | - |

**Bold** significant at p-value less than 0.05.

of coverage in Zambia [41]. This may be explained by differences in the levels of gross domestic product (GDP), while Uganda is in the low income category, both Kenya and Zambia have attained middle income status according to the World Bank rankings [42]. It is expected that as a countries GDP increases, the socioeconomic status of its population also improves, and this has been strongly linked to access and utilisation of maternal healthcare services [43]. Other studies from the region that did not use a nationally representative samples, but reported similar results include; 10% level of coverage for Tanzania in the Morogoro region [37], 38% level of coverage for Western Kenya [44], 8% level of coverage for Ghana from three health demographic surveillance sites of Dodowa, Kintampo, and Navrongo [36] and 7% level of coverage for Lao PDR from a survey in rural Khammoune region [14]. These variations may be due to regional differences in socioeconomics and access to health services.

The level of continued care reduced at every stage of care, largest dropout of 54 percentage points occurred between the stages of childbirth and the postnatal period. Our finding contrasts results from a similar studies done in Ethiopia and Zambia where the dropout rate

between childbirth and the postnatal period was less than 10 and 24 percentage points respectively [35, 41]. The drop in utilisation between childbirth and postnatal care could be a pointer to the quality of service received during childbirth and labour. Prior experience of service received in a health facility has been shown to be a major determinant of future use [45, 46]. The quality of maternal care provided in Uganda has been reported to be wanting across the entire continuum of maternal care in a number of studies [47–50]. The current study did not focus the quality aspects of maternal care offered, but it is an important aspect that needs further investigation by the government because it might help to improve the level of completion of the continuum of maternal care. Additionally, about a quarter of the women deliver at home in rural areas, so it is possible that these may erroneously perceive themselves as not needing a medical check because they are out of danger after the delivery [29, 51]. This is worsened by the fact that community based health services in Uganda do not prioritize PNC hence women who miss returning to health facilities completely lose out [52]. Furthermore, the overburdened and burned-out healthcare are most likely unable to provide satisfactory ANC and intrapartum counselling sessions[53–55]. Unsatisfactory MCH services characterized by long waiting time, limited availability of medical supplies, and poor health workers' attitudes might partly contribute to low PNC utilization as women not motivated to return to health facilities because of poor client-provider relations [48, 54, 55]. It is also possible that the way the survey question is phrased may not be clear enough for many women to understand. ("...*Did anyone check on baby's health after you left place of birth*?" [51]. The immediate postpartum period carries the highest risk for death [7, 56]. Therefore, every parturient and newborn should be assessed by a skilled service provider for early detection and treatment of complications to promote maternal and perinatal survival.

Women who initiated ANC visits in the first trimester were more likely to have completed the continuum of maternal care compared to those who initiated ANC visits after the first trimester. This association between the timing of ANC initiation and the completion of the continuum of maternal care has been consistently demonstrated in previous studies [14, 37, 38]. Early initiation of ANC increases the likelihood of having more ANC contacts, which offers more opportunities and time to attend the health education sessions about the benefits of being cared for by a skilled provider [57]. Additionally, repeated contact between the skilled provider and the client (pregnant woman) helps to build rapport that may promote the likelihood of using the other components of the continuum of maternal care [58]. Good rapport between a provider and client helps to build trust and confidence in the services received, which has been shown to influence the choices women make regarding the place of delivery and uptake of other reproductive services such as use of modern contraceptives [59]. Ongoing health education and empowerment of women by promoting girl child education may help dispel some of the cultural beliefs and myths associated with late initiation of ANC [60–62]. Our results show an increase of 17 percentage points between the attendance of four or more ANC visits and health facility delivery. In the health sector, the decentralization policy has greatly improved physical access because it is estimated that 72% of the population lives within five kilometers of a health facility [28]. This could partly explain the observed increase in the utilisation of all maternal healthcare services such as antenatal care and health facility deliveries because the basic emergency obstetric care services have been brought closer to the community.

Women in Central region were more likely to complete the continuum of maternal care compared to those in the Eastern region while those in Western Uganda were less likely to complete continuum of maternal care. Central Uganda is the commercial and administrative region of the country. Therefore, it is generally well serviced with better social amenities such as schools, health care facilities, mass media and roads than the rest of the country [63]. As

such, one would expect the residents to be of a higher social economic status and with better education compared to the rest of the country. All these factors are known to promote the utilisation of reproductive, maternal, and neonatal child health services because they minimize delays in accessing care [21, 28, 29]. In contrast, women resident in Western Uganda were least likely to complete the continuum of maternal care. This finding needs further investigation because this is not the poorest region of the country and its MCH indicators are comparable to those of the central region [63]. However, this finding may partly be explained by the unique cultural practices and beliefs about pregnancy and childbirth in this region [51, 59]. For instance, anecdotal evidence from this region suggests that for the first seven days after childbirth, the baby should not be seen by strangers. So, babies born at home are kept indoors until the first week of life has elapsed. While those that give birth normally in health facility are in a rush to get back home early to full fill this cultural practice because the health facility does not offer adequate privacy. Currently, Eastern Uganda, is the poorest region in the country with some of the worst MCH indicators [63]. This tends to attract more development partners in the region to support the provision of social services such as maternal care.

Women who had attained at least secondary level of education were more likely to complete the continuum of maternal care compared to those that had only primary level of education. This finding is consistent with reports from previous studies, a higher level of education has been associated with increased likelihood of utilisation of reproductive, maternal, and neonatal child health services [11, 36, 38]. An educated person is able to make better choices regarding one's health and more likely to have financial to pay for health care so they do not have to rely on their husbands to seek care [64]. Therefore, the government of Uganda needs to invest more in education for all, with special attention to the girl child as a long-term strategy to improve the level of utilisation of the continuum of maternal care for improved maternal health.

Exposure to mass media (radio and newspapers or magazines) was positively associated with completion of the continuum of maternal care. Media is a very powerful tool because of its potential to reach many people in a very short time and at a relatively lower cost [65]. This finding is consistent with results from previous studies conducted in different settings [66–68]. Information about the availability of services and their benefits to the public can be quickly disseminated to bridge knowledge gaps and dispel any myths and misconceptions that may exist. The government should leverage the advantages of mass media, to positively influence health-seeking behavior of the Ugandan women. This will improve the level of utilisation of the continuum of maternal care for better MCH in the country. The current UDHS data set did not look at the influence of social media, but since the last census in 2016, social media usage seems to have overtaken the traditional media in coverage. So, it would be an interesting variable to add in future surveys because of the enormous advantages it carries over the traditional media [65]. In line with earlier studies [69], our current analysis shows that there are variations in the influence of different types of mass media, with radio and television being more effective and the policy makers may need to pay attention it [67]. Much as this analysis of secondary data did not reveal any association between socioeconomic status and the complete continuum of maternal care, exposure to mass media may be interpreted as a proxy indicator for women more likely to be of a higher social economic status and more likely to use maternal services [70].

## Study strengths and limitations

This study used a nationally representative sample to explore the level and determinants of the continuum of maternal care for maternal health, so the findings can be easily generalised to all

women in Uganda. Therefore, it provides baseline information that can be used to inform the design of future studies as well as programs to improve maternal and newborn survival. On the other hand, because of the cross-sectional nature of the data, we can only establish associations and not causality. Recall bias is the major threat to validity in such studies because they are based on self-reported responses. DHS dataset lacks detailed variables on PNC such as frequency of PNC which limited our outcome. Women who did not survive their pregnancy or childbirth were not included, neither were pregnancies resulting in stillbirth [71]. Finally, data on some important factors known to influence utilisation of reproductive health services such as access to comprehensive emergency obstetric and neonatal care services were not captured in the UDHS.

## Conclusion

In Uganda, about 11% of women completed the continuum of maternal care, with many discontinuities at different stages. The largest drop occurred between the stages of childbirth and utilisation of PNC services. The factors associated with completion of the continuum of maternal care were early initiation of ANC, use of modern contraceptives, exposure to mass media, level of formal education attained and being a resident of the central and western region. We recommend that the Ministry of Health should leverage the enormous potential of mass media to promote the utilisation of maternal care services. In addition, promoting early initiation of ANC and the integration of MCH services may go a long way in improving continuum of care. The current universal access education programs need to be supported as long-term strategy towards improved MCH. Further studies are needed to understand why most women are unable to return their newborns for PNC after discharge from the places of childbirth.

## Supporting information

**S1 Table. Trends in continuum of care utilization in Uganda over the last two decades.**
(DOCX)

**S2 Table. Predictors of complete continuum of maternity care in Uganda.**
(DOCX)

## Acknowledgments

We thank the MEASURE DHS program for availing us with the data set.

## Author Contributions

**Conceptualization:** Quraish Sserwanja, David Mukunya, Prossy Nabachenje, Julius N. Wandabwa, Milton W. Musaba.

**Data curation:** Milton W. Musaba.

**Formal analysis:** Quraish Sserwanja, David Mukunya, Milton W. Musaba.

**Investigation:** Milton W. Musaba.

**Methodology:** Quraish Sserwanja, David Mukunya, Prossy Nabachenje, Alleluyah Kemigisa, Paul Kiondo, Julius N. Wandabwa, Milton W. Musaba.

**Resources:** Julius N. Wandabwa.

**Supervision:** Paul Kiondo, Julius N. Wandabwa.

**Visualization:** Paul Kiondo.

**Writing – original draft:** David Mukunya, Julius N. Wandabwa, Milton W. Musaba.

**Writing – review & editing:** Quraish Sserwanja, David Mukunya, Prossy Nabachenje, Alle-
luyah Kemigisa, Paul Kiondo, Julius N. Wandabwa, Milton W. Musaba.

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
