## [Decision Letter · Decision Letter 0]

18 Jun 2021

PONE-D-21-14873

Continuum of maternity care for maternal and newborn health in Uganda: a national cross-sectional study.

PLOS ONE

Dear Dr. Milton W. Musaba,

Thank you for submitting your manuscript to PLOS ONE. After careful consideration, we feel that it has merit but does not fully meet PLOS ONE’s publication criteria as it currently stands. Therefore, we invite you to submit a revised version of the manuscript that addresses the points raised during the review process.

Three reviewers have made comments on this manuscript. Though it may take time to revise it, please try to improve your manuscript. New articles are increasing for this topic, and you may continue to update your references.

We look forward to receiving your revised manuscript.

Kind regards,

Masamine Jimba

Academic Editor

PLOS ONE

Journal Requirements:

2. Please upload a copy of Supporting Information S1 which you refer to in your text on page 17.

Reviewers' comments:

Reviewer's Responses to Questions

**Comments to the Author**

1. Is the manuscript technically sound, and do the data support the conclusions?

Reviewer #1: Yes

Reviewer #2: Yes

Reviewer #3: Yes

2. Has the statistical analysis been performed appropriately and rigorously? 

Reviewer #1: Yes

Reviewer #2: Yes

Reviewer #3: Yes

3. Have the authors made all data underlying the findings in their manuscript fully available?

Reviewer #1: Yes

Reviewer #2: Yes

Reviewer #3: Yes

4. Is the manuscript presented in an intelligible fashion and written in standard English?

Reviewer #1: Yes

Reviewer #2: Yes

Reviewer #3: Yes

5. Review Comments to the Author

Reviewer #1: Thank you for giving me a chance to review your manuscript. The manuscript shows the latest status of continuum of care including antenatal care visit, place of child birth, and postnatal care visit in Uganda very clearly. However, I have two major and several minor suggestions as follow. I hope my comments will help to improve the manuscript.

[Major suggestions]

1. I understand that many previous studies measured the complete continuum of care with the same definition used in this study, and DHS collects data on four antenatal care visits and postnatal check within 6-week postpartum. However, WHO recommends at least eight times of antenatal care visits, and four times of postnatal care visits. If the national guidelines in Uganda apply the WHO’s recommendation, this study should measure continuum of care based on the guidelines used at the time of survey.

2. As reading the second paragraph of the discussion section, I thought that readers would like to see the change in antenatal care visits, place of childbirth, postnatal care visit, and continuum of care over years. Because Uganda DHS 2000-01, 2006, and 2011 DHS are available, I would like to encourage authors to add these results, which will show the effect of centralization policy and socio-economic development on utilization of maternity care services in the country.

[Minor suggestions]

<abstract>

1. Please correct “the latest Uganda Demographic and Health Survey” to “Uganda Demographic and Health Survey 2016”.

2. Please report AOR and 95% with the following way: (AOR, 95%CI)

<introduction>

3. In the first paragraph. I wonder the references #10. #11, #12, and #13 provide any evidence that complete continuum of care prevented or reduced maternal and newborn deaths.

4. In the second paragraph, please add the information on how many times and when mothers and their newborns should receive postnatal checkups in the national guidelines in Uganda.

5. In the third paragraph, please show the factors associated with each of the maternal child health care services in Uganda that were reported by previous studies.

[Materials and Methods]

6. In the study setting section, please provide more detail about the situation in Uganda, including maternal and child health indicators, health system, or the characteristic of each region.

<results>

7. Please correct the sample size of 10125 to 10152.

8. Please report the frequency of antenatal care visits, followed by place of childbirth and postnatal care.

9. On Table 2, please report the percentage of complete CoC for each category by row. For example, the percentage of complete CoC among women aged 15-24 by row is 10.9% (385/3546).

10. Please provide the sample size for the adjusted model I and II, respectively.

<discussion]>

11. In the first paragraph, please report percentage only, and delete the values of numerator and denominator indicated after the percentage.

12. In the fourth paragraph, authors mentioned that Western Uganda has unique cultural practice and belief about pregnancy and childbirth. Please explain more details.

.</discussion]></results></introduction></abstract>

Reviewer #2: Overall, the authors discussed the important topic of the continuum of care for maternal and newborn health in Uganda. They conducted a secondary analysis of the 2016 Demographic and Health Survey to determine the prevalence and factors associated with the continuum of maternity care in Uganda. There was a general lack of interpretation of results regarding the discontinuities between antenatal care, health facility childbirth, and postnatal care services. The associated factors, such as initiation of ANC in the first trimester, level of education, and exposure to mass media, have been reported in several studies. Therefore, it would be great if the authors could add evidence and implication for policy makers based on findings of this study.

Please find my comments in blue in the attachment.

Reviewer #3: The authors analyzed the level of continuum of care in maternal, newborn, and child health in Uganda using a nationally representative data. The manuscript is easy to read, and the analysis is conducted with a widely-accepted method. The reviewer hopes that the following comments help the authors to improve the manuscript.

Major comment

1. When did the Government of Uganda started recommending ANC eight times? The authors explained that “In conformity with the current World Health Organisation (WHO) guidelines, the Ministry of Health in Uganda recommends that all women attend at least eight ANC visits and have childbirth in a health facility supervised by a skilled service provider [16].” (in Page 2). However, since the data were collected in 2016, it is important to mention the guidelines in Uganda when the survey was conducted. In addition, this statement is not consistent with ANC component (ANC 4+) under the definition of continuum of care completion.

2. What are the recommended timings of PNC in Uganda? Since it is related to how the continuum of care should be defined, tt is worth mentioned in the Introduction section.

3. Why did the author define the PNC component of the continuum of care completion as “at least one postnatal check-up within six weeks after childbirth?” A major limitation of a continuum of care analysis using DHS dataset is lack of detailed variables on PNC. If this definition is not consistent with the PNC guidelines in Uganda, the authors may want to add this inconsistency as a limitation.

4. Is the variable for PNC based on PNC for mothers or PNC for children? Typically, DHS dataset has both of the variables. The authors need to explain the definition of the PNC component of the continuum of care completion clearly.

5. The authors mentioned that “The UDHS interviewed a total of 18,506 women aged 15–49 years” and “Our secondary analysis was a total enumeration of all the 10,152 women who had had a live birth within five years preceding the survey [23].” (in Page 3). And the authors analyzed 10,125 women according to the description in the Results section. The authors need to clarify why the number of women was limited to 10,125 from 18,506 or 10,152, either in the Methods or Results section.

6. Eligible women included in the analysis seem to be different by indicators: complete continuum-of-care (10,125 women), four ANC or more (7,262), facility delivery (10,125), and (10,125 women). The column heading of Table 1 stated that N=10,152 and rows for place of delivery, postnatal care, anc visits, and complete continuum of care had 10,152 observations. Please check if these data are correctly analyzed and reported.

7. How many women were included in the logistic regression analysis? The adjusted model II included the variables with missing observations, namely, husbands’ education level and care seeking decisions. Therefore, the number of women included in the adjusted models I and II may differ. Please clarify the number of women in the column heading of Table 2 (or any other appropriate place).

8. Why did the Region variable have 4 categories only? According to DHS 2016 report (https://dhsprogram.com/pubs/pdf/FR333/FR333.pdf), there are 15 Regions. Although the continuum of care may be affected by supply-side characteristics, DHS dataset does not have such characteristics. A geographic area variable, such as the Region variable, can capture external factors that cannot be controlled by women’s and households’ characteristics in the regression model, including socioeconomic and healthcare service levels at the area level. It would be recommended not to merge the Region variable. Note that previous papers often use binary variables that capture geographic areas or multilevel models at the area level.

Minor comment

1. In the Abstract, please clarify the survey year (2016).

2. Introduction (Page 2): Data in the first paragraph (such as 277,300 maternal deaths, 1 in 59 160, or 5,400 pregnancies; 27 deaths per 1,000 livebirths, …) should be introduced with the year(s) of survey.

3. In Table 2, the superscript “a” for the variable of “Husband’s education level” does not have a corresponding footnote.

4. In the first paragraph of the Discussion section, “54%” in the sentence, “The largest dropout of 54% occurred between the stages of childbirth and the postnatal period.” might have been 54 percentage points (76.6% - 22.5%). Note that 22.5%/76.6%=0.294 and it could be presented “71% dropout.”

5. In the second paragraph of the Discussion section, “16%” in the sentence, “Our results show an increase of 16% between the attendance of four or more ANC visits and health facility delivery,” might have been 17 percentage points (76.6% - 59.9%). Note that 76.6%/59.9%=1.28 and it could be presented “28% difference.”

6. PLOS authors have the option to publish the peer review history of their article (what does this mean?). If published, this will include your full peer review and any attached files.

Reviewer #1: No

Reviewer #2: No

Reviewer #3: **Yes: **Akira Shibanuma

---

## [Author Response · Author response to Decision Letter 0]

26 Jul 2021

24th/11/2019

To 

Dr. Masamine Jimba

Academic Editor

PLOS ONE

Dear Dr. Masamine Jimba

Re; Response to reviewers’ comments and resubmission of revised manuscript PONE-D-21-14873

Thank you for taking off time to review and provide feedback on this manuscript titled “Continuum of maternity care for maternal and newborn health in Uganda: a national cross-sectional study”. Please receive the revised manuscript and a point-by-point response to the comments raised by the peer reviewers as summarized in the table below. 

Comment Response Line

Reviewer #1

Thank you for giving me a chance to review your manuscript. The manuscript shows the latest status of continuum of care including antenatal care visit, place of child birth, and postnatal care visit in Uganda very clearly. However, I have two major and several minor suggestions as follow. I hope my comments will help to improve the manuscript. We are grateful for the time you dedicated to provide feedback to improve this manuscript. NA

Major suggestions

1. I understand that many previous studies measured the complete continuum of care with the same definition used in this study, and DHS collects data on four antenatal care visits and postnatal check within 6-week postpartum. However, WHO recommends at least eight times of antenatal care visits, and four times of postnatal care visits. If the national guidelines in Uganda apply the WHO’s recommendation, this study should measure continuum of care based on the guidelines used at the time of survey. Thank you so much for the observation. This survey was conducted between 2011 and 2016 (five years preceding the survey) yet the WHO guidelines of at least 8 times was introduced globally in 2016. Relatedly, the MoH in Uganda has just changed its ANC guidelines in the last two years towards the eight ANC contacts. So, we decided to use the guidelines that were being implemented at the time of the survey. Goal Oriented Antenatal Care Protocol | Ministry of Health Knowledge Management Portal. This information has been provided under the sections on study setting and independent variables. 119 and 159

2. As reading the second paragraph of the discussion section, I thought that readers would like to see the change in antenatal care visits, place of childbirth, postnatal care visit, and continuum of care over years. Because Uganda DHS 2000-01, 2006, and 2011 DHS are available, I would like to encourage authors to add these results, which will show the effect of centralization policy and socio-economic development on utilization of maternity care services in the country. Thank you for the observation. This is something that we had done in the analysis but decided not to include in this manuscript because there were a lot of inconsistencies in the data. However, we have now included this information in a supplementary table 1 to show the trends. Although, we think this does not show the effect of centralization policy and socio-economic development on utilisation of maternity care services in the country as the reviewer suggests. 227-228

Minor suggestions

1. Please correct “the latest Uganda Demographic and Health Survey” to “Uganda Demographic and Health Survey 2016”. This has been corrected in the methods section of abstract. 21

2. Please report AOR and 95% with the following way: (AOR, 95%CI) This has been corrected 29 -37

164 -174

3. In the first paragraph. I wonder the references #10. #11, #12, and #13 provide any evidence that complete continuum of care prevented or reduced maternal and newborn deaths. Thank you for this query, these papers review the literature on the rational for and importance of the continuum of care in reducing maternal and perinatal morbidity and mortality. Although, they all aim to identify factors associated with completion of the continuum of maternity care 56 - 60

4. In the second paragraph, please add the information on how many times and when mothers and their newborns should receive postnatal checkups in the national guidelines in Uganda. The following statement has been added “For home births, the first postnatal check should occur within 24 hours of birth. Thereafter, all mothers and newborns should receive at least three additional postnatal checks on day three, between day seven and 14, and between week two and six following child birth” 72-74

5. In the third paragraph, please show the factors associated with each of the maternal child health care services in Uganda that were reported by previous studies. These factors have been added to the third paragraph as suggested. 84 - 94

Materials and Methods

6. In the study setting section, please provide more detail about the situation in Uganda, including maternal and child health indicators, health system, or the characteristic of each region. Thank you for the suggestion, these details have been added. 110 - 134

7. Please correct the sample size of 10,125 to 10,152. Sorry for the typo, this has now been corrected. 144

8. Please report the frequency of antenatal care visits, followed by place of childbirth and postnatal care. We reported these in this order both in the abstract and results. 30-31

191-192

9. On Table 2, please report the percentage of complete CoC for each category by row. For example, the percentage of complete CoC among women aged 15-24 by row is 10.9% (385/3546). Thank you for the suggestion. However, we present the preferred column percentages in all our tables 216 - 217

10. Please provide the sample size for the adjusted model I and II, respectively. Added in table 2 216 - 217

11. In the first paragraph, please report percentage only, and delete the values of numerator and denominator indicated after the percentage. This has been done as advised 225-226

12. In the fourth paragraph, authors mentioned that Western Uganda has unique cultural practice and belief about pregnancy and childbirth. Please explain more details. For instance, anecdotal evidence from this region suggests that for the first seven days after childbirth, the baby should not be seen by strangers. So, babies born at home are kept indoors until the first week of life has elapsed. While those that give birth normally in health facility are in a rush to get back home early to full fill this cultural practice because the health facility does not offer adequate privacy. 301-306

Reviewer #2

Overall, the authors discussed the important topic of the continuum of care for maternal and newborn health in Uganda. They conducted a secondary analysis of the 2016 Demographic and Health Survey to determine the prevalence and factors associated with the continuum of maternity care in Uganda. There was a general lack of interpretation of results regarding the discontinuities between antenatal care, health facility childbirth, and postnatal care services. The associated factors, such as initiation of ANC in the first trimester, level of education, and exposure to mass media, have been reported in several studies. Therefore, it would be great if the authors could add evidence and implication for policy makers based on findings of this study. 

Please find my comments in blue in the attachment. Thank you for taking time to review this manuscript and provide comprehensive feedback to make it better. We believe that we have addressed all the issues raised by the reviewers. NA

Abstract

Results: About 59.9% (6080/7262) (95% CI: 59.0-60.8) had four or more antenatal visits. -Please check if the denominator is correct. I think you do not need to report exact numbers every time if you report results with a percentage with the denominator. Thank you for the observation. This was an error that we have corrected. The denominator should be 10,152. Your suggestion has been adopted. 29-32

Introduction

The first paragraph seems too long. It needs to be restructured to provide an overview of maternal, neonatal and under-5 childhood deaths by comparing sub-Saharan Africa with the other regions of the world. We have modified the paragraph accordingly 225-254

Almost all 277,300 (94%) of the annual maternal deaths occur in low and middle-income countries and roughly two-thirds (196,000) of these deaths happen in South East Asia and sub- Saharan Africa [2,3]. - How many annual maternal deaths occurred? When you report percentages, please indicate their denominators together. Thank you for the advice, this has been implemented 45 - 46

It is estimated that 45% of all the under-5 childhood deaths happen early in the neonatal period, especially in the first seven (7) days of life. - Is it the global situation or and sub-Saharan Africa? This is in low- and middle-income countries. 55

In introduction, you need to describe the situations of Uganda in terms of maternal, neonatal and under-5 childhood deaths if available. It is also recommended to include country data related to the study objectives briefly, such as the number of women of reproductive age, ANC and PNC attendance. It would provide rationales for the study objectives and the interventions for the continuum of maternity care recommended in the discussion. Thank you for this suggestion. We have provided this information under the study setting section. 110 - 134

The current study focused on the completion of the continuum of care by specifically looking at antenatal care (ANC) attendance, health facility utilization during childbirth, and postnatal care (PNC). – This sentence in the second paragraph should be moved to the next paragraph as it describes the study objectives. This suggestion is noted and it has been implemented accordingly 98-100

It is better to include evidence available on the association between the continuum of maternity care and maternal and newborn health if available in the second paragraph. This evidence has been included here “It is estimated that 80% of maternal deaths and 66% of neonatal deaths in the world could be averted by ensuring that women have access to quality and effective maternity care services especially during childbirth and in the first week of life [9,10].”

56-58

Materials and Methods

Study design: it should be mentioned that this study is the secondary analysis of the nationally representative cross-sectional study (UDHS) conducted between June and December 2016. This has been added. 141

Sample size and sampling: Did all the sample of 10,152 women have the outcome data? If you excluded some women due to missing data on the outcome, please explain it or add a study population flow chart.

 No missing data on the outcome was noted. However, PNC had 44 participants, ANC 56 who said, “I don’t know” so these were considered as if they had no PNC check and had less than 4 visits, respectively. Place of delivery had no I do not know responses. 143-145

Outcome Variable: Complete continuum of care was the outcome variable. Complete continuum of maternity care was considered when a woman reported having had: at least four ANC (ANC4+) contacts during the most recent pregnancy, utilized a health facility during the most recent childbirth and had at least one postnatal check-up within six weeks after childbirth: It would be great if you could add definitions of each segment consisting continuum of care. Are they in line with the Uganda MOH/WHO’s recommendations, for instance, in terms of the number and quality of ANC visits, supervision of a skilled service provider and length of stay in a health facility during childbirth and timing of postnatal check-up? 1. Until about two years ago, ANC guidelines in Uganda recommended at least 4 ANC visits for every pregnant woman.

2. Each pregnant woman is expected/recommended to give birth under the care of skilled birth attendant 

3. The same guidelines recommend that each woman should have at least four checks in the postnatal period. However, one of the limitations of the UDHS data is that it only asks about only one PNC visit. So, information about the other three visits is not available. 151-157

Independent variables: Some variables included in the analyses and the result tables were not listed here, such as contraceptive use and religion. For variable selection, it is better to refer to a conceptual framework, such as Owilli’s continuum of care framework for maternal, newborn and child health (https://doi.org/10.1186/s12889-016-3075-0). Thank you for the observation and for pointing us to this conceptual frame work to refer to. Variables that were not listed in the methods have now been listed to include those in Owilli’s framework, which we had measured. 172-173

187-188

Results

Characteristics of the study population: If you describe 95% CI in the narrative part, it is better to include them in Table 1 as well. Also, recheck Table 1 as total numbers exceed the sample size for some variables (i.e., postnatal care, residence). Thanks for the observation. This has been corrected in table 1 204-205

Of the 10,125 women, 59.9% (6,080/7262) (95% CI: 59.0-60.8) had four or more ANC contacts while 76.6% (7,780/10,125) (95% CI: 75.8-77.5) utilized health facility for childbirth and 22.5% (2,280/10,125) (95% CI: 21.5-23.2) attended postnatal care within six weeks after childbirth. – Is the denominator for four or more ANC visits correct? Thanks this has been corrected. 195-199

Factors associated with complete continuum of maternity care: The description is too short. You need to explain the key statistical outputs provided in Table 2. Which model is the main model that you used to discuss the results? In the description, were those independent variables associated with the outcome negatively or positively? For this, I think it is important to include odds ratios for some key independent variables. Thank you for the important observation, these have now been provided as advised. 207-217

Throughout the manuscript, there were inconsistencies in reporting formats and typos, such as the use of thousand separators. Please review the whole manuscript. The whole manuscript has been revised to standardize the writing. NA

Discussion

We found a low level of utilization of complete continuum of maternity care at 11%. The level of use was highest for the place of childbirth at (7,789/10,125):77%), followed by ANC 4 or more visits at (6,080/7,262: 60%) and lowest for PNC at (2,280/1, 0125: 23%). - The discussion of the study seems mixed with results. This sentence was already presented in the results. You need to add more interpretation of the findings in the discussion. Thank you for this advice, it has been implemented 229-231

Uganda’s levels of completion of the continuum of maternity care is similar to that of other countries in the region. For both Ethiopia and Tanzania, the reported level is at about 10% [27,29], 8% for Ghana [28] and 7% for Lao [12]. Compared to Kenya, it is more than three times below the reported level of completion of the continuum of maternity care at 36% [32]. - You compared the prevalence of a complete continuum of maternity care among several countries. Did these studies use the same definition of the continuum of maternity as one used for this study? Moreover, are they all national-level studies? Please note the level of continuum of care may vary even within a country. Therefore, if you want to compare the prevalence of continued care with other studies, they should be nationally representative data. Otherwise, please specify locations, for instance, districts and regions. Thank you so much for this important observation. Corrections have been made to reflect these very specific details in the comparisons that have been made. 232-238

The largest dropout of 54% occurred between the stages of childbirth and the postnatal period. - I think it would be useful if you could provide more insights on this point by comparing it with the situations of other counties. For instance, how about the situation in Kenya? This has been done 239-246

In the health sector, the decentralization policy has greatly improved physical access because it is estimated that 72% of the population lives within five kilometers of a health facility [34].This could partly explain the observed increase in the number health facility deliveries because the basic emergency obstetric care services have been brought closer to the community. - This paragraph is confusing. This study used only UDHS 2016 data, so you were not able to explore changes over time. If you want to discuss the increase in the number of health facility deliveries, you need to present the baseline. You did not compare longitudinal data and present any data about health facility delivery in the past in this manuscript. Decentralization policy may have increased all the ANC, health facility deliveries, PNC. This policy was not discussed before, so you need to explain this. This is noted, in the study setting section, we have provided more details on the decentralized nature of health care services in Uganda. You rightly point out that decentralization should ideally increase utilisation of all services (ANC, facility births, PNC). In this regard, we have added a supplementary file to show the changes in maternal child health services over time. 115 -139

Our results show an increase of 16% between the attendance of four or more ANC visits and health facility delivery… On the other hand, the observed drop in utilization between childbirth and postnatal care could be a pointer to the quality of service received during childbirth and labour. - They are directly related to the continued care. It would be useful if you could elaborate on why these discontinuities occurred, apart from the poor quality of maternity care in Uganda. This has been improved, in the discussion section 250-261

Most of the factors associated with the continued care, such as initiation of ANC in the first trimester, level of education, and exposure to mass media, were reported and examined in previous studies. It is great if you could add evidence to them based on the findings of this study. At the same time, you described each associated factor using one paragraph, which is a bit wordy. It is great you can summarize discussions and put more focus on the unique findings of your study. Thank you for this observation, the key finding in this study is the low level of the continuum of care because of the sharp drop between the stages of child birth and utilisation of PNC services. The first two paragraphs in the discussion explore this finding. 272-277

You found large geographic variations in the continuum of maternity care. It is interesting if you can further investigate why the Eastern area, the poorest region with some of the poorest maternal and child health indicators, had a better continuum of care than the Western area. For instance, are there development partners supporting the continuum of care in this region? Thank you for the suggestion, this has been added. 311-312

This finding may partly be explained by the unique cultural practices and beliefs about pregnancy and childbirth in this region – Please provide a few examples here. These have been added

 305 - 310

I think it would be great if you could include recommendations beyond mass media and education programs to promote the complete continuum of care, such as support for early initiation of ANC and the integration of ANC and PNC programs based on your findings. Thank you for the advice, this has been added. 333-334

Reviewer #3

The authors analyzed the level of continuum of care in maternal, newborn, and child health in Uganda using a nationally representative data. The manuscript is easy to read, and the analysis is conducted with a widely-accepted method. The reviewer hopes that the following comments help the authors to improve the manuscript. Thank you very much for these comments NA

Major comment

1. When did the Government of Uganda started recommending ANC eight times? The authors explained that “In conformity with the current World Health Organisation (WHO) guidelines, the Ministry of Health in Uganda recommends that all women attend at least eight ANC visits and have childbirth in a health facility supervised by a skilled service provider [16].” (in Page 2). However, since the data were collected in 2016, it is important to mention the guidelines in Uganda when the survey was conducted. In addition, this statement is not consistent with ANC component (ANC 4+) under the definition of continuum of care completion. Thanks so much for the observation. This day was collected on births that occurred between 2011 and 2016 (five years preceding the survey) yet the WHO guidelines of at least 8 times was introduced globally in 2016. Hence, we decided to use the guidelines at the time when the outcome was occurring. 72-75

2. What are the recommended timings of PNC in Uganda? Since it is related to how the continuum of care should be defined, tt is worth mentioned in the Introduction section. This has been included as recommended. 72-75

3. Why did the author define the PNC component of the continuum of care completion as “at least one postnatal check-up within six weeks after childbirth?” A major limitation of a continuum of care analysis using DHS dataset is lack of detailed variables on PNC. If this definition is not consistent with the PNC guidelines in Uganda, the authors may want to add this inconsistency as a limitation. Thank you for the suggestion, we have added this in the limitation section. 14/315-316

4. Is the variable for PNC based on PNC for mothers or PNC for children? Typically, DHS dataset has both of the variables. The authors need to explain the definition of the PNC component of the continuum of care completion clearly. We used the women's dataset, in which the unit of analysis is the woman so we consider it as woman whose child had a postnatal check 160-161

5. The authors mentioned that “The UDHS interviewed a total of 18,506 women aged 15–49 years” and “Our secondary analysis was a total enumeration of all the 10,152 women who had had a live birth within five years preceding the survey [23].” (in Page 3). And the authors analyzed 10,125 women according to the description in the Results section. The authors need to clarify why the number of women was limited to 10,125 from 18,506 or 10,152, either in the Methods or Results section. Thank you for the observation. It was an error on our side to write 10,125 instead of 10, 152. We analyzed data of all 10,152 women who had a live birth 5 years preceding the survey. 149

6. Eligible women included in the analysis seem to be different by indicators: complete continuum-of-care (10,125 women), four ANC or more (7,262), facility delivery (10,125), and (10,125 women). The column heading of Table 1 stated that N=10,152 and rows for place of delivery, postnatal care, anc visits, and complete continuum of care had 10,152 observations. Please check if these data are correctly analyzed and reported. Thanks for the observation. It was an error on our side to write 10,125 instead of 10, 152. We analyzed data of 10,152 women who had a live birth 5 years preceding the survey. This has been corrected. 149-150

221-222

7. How many women were included in the logistic regression analysis? The adjusted model II included the variables with missing observations, namely, husbands’ education level and care seeking decisions. Therefore, the number of women included in the adjusted models I and II may differ. Please clarify the number of women in the column heading of Table 2 (or any other appropriate place). This has now been clarified in Table 2. 221-222

8. Why did the Region variable have 4 categories only? According to DHS 2016 report (https://dhsprogram.com/pubs/pdf/FR333/FR333.pdf), there are 15 Regions. Although the continuum of care may be affected by supply-side characteristics, DHS dataset does not have such characteristics. A geographic area variable, such as the Region variable, can capture external factors that cannot be controlled by women’s and households’ characteristics in the regression model, including socioeconomic and healthcare service levels at the area level. It would be recommended not to merge the Region variable. Note that previous papers often use binary variables that capture geographic areas or multilevel models at the area level. Thanks so much for the observation. The UDHS used sub-regions which were 15. However, in our analysis, we used region as a variable so we used the four regions of Uganda (Northern, Eastern, Central and Western) by combining the different sub-regions that make up regions. 

Northern (Teso, Karamoja, Lango, Acholi, West Nile), Central (Kampala, Central 1 and Central 2), Eastern (Busoga, Bugishu and Bukedi) and Western (Tooro, Ankole, Bunyoro and Kigezi)

The same has also been used in other papers 

(e.g Yang YY, Kaddu G, Ngendahimana D, Barkoukis H, Freedman D, Lubaale YA, Mupere E, Bakaki PM: Trends and determinants of stunting among under-5s: evidence from the 1995, 2001, 2006 and 2011 Uganda Demographic and Health Surveys. Public health nutrition 2018, 21(16):2915-2928). 173 - 174

Minor comment

1. In the Abstract, please clarify the survey year (2016). This has been done 21

2. Introduction (Page 2): Data in the first paragraph (such as 277,300 maternal deaths, 1 in 59 160, or 5,400 pregnancies; 27 deaths per 1,000 livebirths, …) should be introduced with the year(s) of survey. The suggestion has been taken and implemented accordingly 45-51

3. In Table 2, the superscript “a” for the variable of “Husband’s education level” does not have a corresponding footnote. Sorry for the error, this has been corrected 221-222

4. In the first paragraph of the Discussion section, “54%” in the sentence, “The largest dropout of 54% occurred between the stages of childbirth and the postnatal period.” might have been 54 percentage points (76.6% - 22.5%). Note that 2.5%/76.6%=0.294 and it could be presented “71% dropout.” Thank you for this brilliant observation, we have corrected this 244

5. In the second paragraph of the Discussion section, “16%” in the sentence, “Our results show an increase of 16% between the attendance of four or more ANC visits and health facility delivery,” might have been 17 percentage points (76.6% - 59.9%). Note that 76.6%/59.9%=1.28 and it could be presented “28% difference.” Thank you for this brilliant observation, we have corrected this. 262

END

---

## [Decision Letter · Decision Letter 1]

25 Aug 2021

PONE-D-21-14873R1

Continuum of maternity care for maternal and newborn health in Uganda: a national cross-sectional study.

PLOS ONE

Dear Dr. Musaba,

Thank you for submitting your manuscript to PLOS ONE. After careful consideration, we feel that it has merit but does not fully meet PLOS ONE’s publication criteria as it currently stands. Therefore, we invite you to submit a revised version of the manuscript that addresses the points raised during the review process.

We look forward to receiving your revised manuscript.

Kind regards,

Masamine Jimba

Academic Editor

PLOS ONE

Additional Editor Comments:

Thank you for your submission. Previous three reviewers asked you to revise many basic points and you revised most of them. This time, I would like to add my comments in addition to other reviewers’ further comments.

1. L2: Your title includes “newborn health,” but it is not well taken up in the manuscript. Please consider to remove it.

2. L18 and other lines: In many articles about the continuum of care in MNCH, “coverage” is used instead of “prevalence.” Please consider to use “coverage.”

3. L23: In this manuscript, “continuum of care” and “continuum of maternal care” seem to be used interchangeably, which makes readers very confusing. Please be careful about using both phrases.

4. L27: …“continuum of maternal care” or “continuum of care”?

5. L32: ...attended “at least one” postnatal care…

6. L32: “A is associated with B” is not equal with “B is associated with A”. It might be better to write “The following factors were associated with the completion of the continuum of care” or …”…continuum of maternal care”: They were secondary level of education…

7. L35: What do you mean by “print”? Newspapers and magazines? Or either one?

8. L40: Out of many factors associated with the continuum of care or continuum of maternal care, only some factors were written in here. Are they because they were easily modifiable? Simple repetition of results are not recommendable in the conclusion section.

9. L102: Spell out UDHS when it first appears.

10. L104: No need to use abbreviation such as UBOS, ICFPNFPs, PFPs, TCMPs as these abbreviations are not used after this.

11. L119: Instead of “Generally,” you may write “In Uganda,”

12. L136: …(UDHS)? Any way to rewrite it?

13. L136 …secondary analysis or secondary data analysis?

14. L139: In our secondary data analysis, we used data that were obtained from validated women’s… It is better to use “we” instead of using passive tense in the method section. Please recheck your whole method section from L139 to L189.

15. L152: Again, “continuum of maternal care” or “continuum of care”?

16. L163: …exposure to newspapers/magazines,…

17. L179: …proportions…Use of capital letters is not appropriate.

18. L196, 197, 198: 95% CI can be cut as it is available from the Table1.

19. L202: Table 1: Percent (%) and 95% CI?

20. L207-L215: …“continuum of maternal care” or “continuum of care”?

21. L226: …“continuum of maternal care” or “continuum of care”?

22. L227: In this context, 10.7% is better.

23. L225-237: The coverage of the continuum of care in different countries followed the same definition? Are they comparable? All of them were measuring “continuum of maternal care” or “continuum of care”?

24. L232: 10.0% in Tanzania?

25. L233: 8.0% in Ghana?

26. L234: 7.0% in Lao PDR?

27. L234: Lao should be Lao PDR.

28. L237: 36.0%?

29. L250 to 255: This paragraph is well written. It is optional, but it might be better to write a recommendation not in each paragraph but at the end. The current one is somehow acceptable, though.

30. L264-268: This sentence is redundant and it is better not to use a word ‘fact’ in this context.

31. L275: A new phrase appears in here, “continued care.” Please use the same expression.

32. L307-309: This topic sentence does not seem to make sense. Please discuss your data, instead of discussing ref 8, 38 and 40.

33. L312: …“don’t” should be “do not”.

34. L316: …radio or TV? Or both? In the results, it was TV only.

35. L312: Are you discussing maternal and newborn health or only maternal health?

36. L329: What do you mean by print? Newspapers and magazines? Or either one?

37. L329: …television or TV?

38. L329: …“continuum of maternal care” or “continuum of care” or “continuum of care for maternal and newborn health”?

39. L331: secondary analysis or secondary data analysis? The nuance is different.

40. L335: You mean study limitations? This should be a separate subheading like “Conclusion.”

41. L347-8: … “continuum of maternal care” or “continuum of care” or “continuum of care for maternal and newborn health”?

42. L351: “being a resident of western region was positively…” This is misleading.

43. L352: …”ministry of health” or “Ministry of Health”?

44. L352: Low levels of what kind of education? Why only northern and eastern regions?

45. L355-356: This is not a complete English sentence.

46. L356-357: Is this relevant as a conclusion of this article?

47. Reference section should be checked one by one very carefully. For example the journal name of the reference 3 is wrong. When you use PDF files as references, please add access dates. Please do not rely on computer soft wares for making the reference section.

Reviewers' comments:

Reviewer's Responses to Questions

**Comments to the Author**

1. If the authors have adequately addressed your comments raised in a previous round of review and you feel that this manuscript is now acceptable for publication, you may indicate that here to bypass the “Comments to the Author” section, enter your conflict of interest statement in the “Confidential to Editor” section, and submit your "Accept" recommendation.

Reviewer #1: All comments have been addressed

Reviewer #2: All comments have been addressed

Reviewer #3: All comments have been addressed

2. Is the manuscript technically sound, and do the data support the conclusions?

Reviewer #1: Partly

Reviewer #2: Yes

Reviewer #3: Yes

3. Has the statistical analysis been performed appropriately and rigorously? 

Reviewer #1: Yes

Reviewer #2: Yes

Reviewer #3: Yes

4. Have the authors made all data underlying the findings in their manuscript fully available?

Reviewer #1: Yes

Reviewer #2: Yes

Reviewer #3: Yes

5. Is the manuscript presented in an intelligible fashion and written in standard English?

Reviewer #1: No

Reviewer #2: Yes

Reviewer #3: Yes

6. Review Comments to the Author

Reviewer #1: Thank you for responding to my suggestions, and revising the manuscript.

I saw the manuscript has been improved. However, I would to like to offer further suggestions to the authors.

The descriptive result of place of childbirth, postnatal care, ANC visits, and complete continuum of care are the main finding of the study. I suggest the authors to make another table of these results to highlight them. Additionally, please show the results in the table with the following order: ANC visits, place of childbirth, postnatal care, and complete CoC. I found the data on ANC visits 4 and above is missed in Table 1. It is interesting to see the percentage of postnatal care among women delivered at facility and home, respectively, because the place of birth would affect the attendance of postnatal care.

I saw that the authors preferred to presenting percentage by column in Table. I, which I accepted. If presenting the percentage by column, it will be better to show the percentage of each category for women who did not complete CoC too, because readers are interested in difference in the distributions between women who completed and did not complete CoC.

Thank you for adding supplementary table 1. This is informative. Please briefly explain the calculations in the methods, results, and discussion sections. What do 56.3, 4.7, 44.2, and 10.9 mean in the cells of PNC 2001 and 2006. Why some results are reported with 95% CIs, and some are not. Please check missing values and clean the table.

Line23, Line105: Regarding sampling methods, the authors mentioned multistage stratified sampling in the abstract and stratified two-stage cluster sampling in the methods section. Please use one.

Line35, Line329: Please change print to newspaper.

Line81: Delete a double quotation.

Line111: Please spell out 7.3M.

Line115-116: Please use a reference to explain improved access to emergency obstetric and neonatal care services.

Line253: Please revise “every parturient should be …” to “every parturient and newborn should be…”

Line278: The effect of early initiation of ANC is widely known. I suggest the authors to discuss more about why women do not attend the first ANC in the first trimester, and potential interventions to improve the situation.

Reviewer #2: Dear authors

Thank you for addressing the reviewer’s comments. The revised paper is much improved, and the authors' responsiveness is much appreciated.

I have only a few more comments:

1. Please mention when the MoH Uganda started recommending at least eight ANC visit in the introduction (lines 64-67). Also, explain the guideline on ANC visits when the DHS data were collected, which recommended at least four ANC visit, in either study settings or outcome variables.

2. The current manuscript requires a detailed review of formatting. Please spell out abbreviations only when they are used for the first time (e.g., ANC, PNC)

3. In “characteristics of the study population” in the results, I think you do not need to report exact numbers every time if you report percentages with their denominators, considering the word count. Please consider deleting exact numbers (i.e., 7,780/10,152).

4. The “Study setting” seems a bit too lengthy (lines 101-134). The authors are encouraged to revise the subsection focusing on key information related to the continuum of care for maternal and newborn health in Uganda.

Reviewer #3: The authors modified the manuscript by addressing errors and added explanations and interpretations. However, please note that the mark-up copy of the revised manuscript attached to the submission file was not fully consistent with the clean version, which made difficult for review the revisions. In the Response-to-reviewers file, the line numbers for some point-to-point responses may not be correct.

1. In the current submission, the authors presented the prevalence of individual components of services, namely ANC, place of delivery, and PNC, in Supplementary Table 1. Since the authors used these results extensively in the Discussion section, the authors seem to consider these results as important. The authors may want to describe the results of this supplementary table in the Results section.

2. The authors seem to report the multivariable logistic regression analysis results of this study using Model II (N=8044) excluding those without currently married, not model I that included every eligible participant (N=10152). While the two models reported largely consistent results, the authors may want to use model I as the main result as it did not have loss of participants. They may use model II results as supplements (it simply shows that husbands’ education was not statistically significant and the results of multivariable regression analysis were not affected substantially after including husbands’ education and excluding participants without currently married.

3. (minor reporting issue) The authors stated that “Our results show an increase of 16 percentage points between the attendance of four or more ANC visits and health facility delivery.” in the Discussion section (Lines 256-). The revised manuscript has Supplementary Table 1, which reported the figures at the decimal point. The coverage of ANC four times or more was 59.9% and that of health facility delivery was 76.6%. The difference may be 16.7 percentage points. The sentence above may be “… an increase of 17 percentage points….” Please check it.

7. PLOS authors have the option to publish the peer review history of their article (what does this mean?). If published, this will include your full peer review and any attached files.

Reviewer #1: No

Reviewer #2: No

Reviewer #3: **Yes: **Akira Shibanuma

---

## [Author Response · Author response to Decision Letter 1]

29 Sep 2021

18th /09/2021

To 

Dr. Masamine Jimba

Academic Editor

PLOS ONE

Dear Dr. Masamine Jimba

Re; Response to reviewers’ comments and resubmission of revised manuscript PONE-D-21-14873

Thank you for taking off time to review and provide feedback on this manuscript titled “Continuum of maternity care for maternal health in Uganda: a national cross-sectional study”. Please receive the revised manuscript and a point-by-point response to the comments raised by the peer reviewers as summarized in the table below. 

Comment Response Line

Additional Editor Comments

Thank you for your submission. Previous three reviewers asked you to revise many basic points and you revised most of them. This time, I would like to add my comments in addition to other reviewers’ further comments. We are grateful for the time you dedicated to provide feedback to improve this manuscript. NA

1. L2: Your title includes “newborn health,” but it is not well taken up in the manuscript. Please consider to remove it. The words “ and newborn” have been removed 2,101,333

2. L18 and other lines: In many articles about the continuum of care in MNCH, “coverage” is used instead of “prevalence.” Please consider to use “coverage.” Thank you for the suggestion, we have replaced the word prevalence with coverage in the whole manuscript. 19,29,39,96 and 115

3. L23: In this manuscript, “continuum of care” and “continuum of maternal care” seem to be used interchangeably, which makes readers very confusing. Please be careful about using both phrases. Sorry about this confusion, we have chosen to use “Continuum of maternity care” throughout the document. Whole document

4. L27: …“continuum of maternal care” or “continuum of care”? We have chosen to use “Continuum of maternity care” throughout the document. Whole document

5. L32: ...attended “at least one” postnatal care… Thank “at least one postnatal care visit” has been added 34

6. L32: “A is associated with B” is not equal with “B is associated with A”. It might be better to write “The following factors were associated with the completion of the continuum of care” or …”…continuum of maternal care”: They were secondary level of education… Thank you for the advice, this has been revised accordingly. 35 - 40

7. L35: What do you mean by “print”? Newspapers and magazines? Or either one? This has been clarified, it means both Newspapers and magazines 39

8. L40: Out of many factors associated with the continuum of care or continuum of maternal care, only some factors were written in here. Are they because they were easily modifiable? Simple repetition of results are not recommendable in the conclusion section. Yes, we chose the easily modifiable factors. This has now been stated more explicitly. 42 - 47

9. L102: Spell out UDHS when it first appears. This has been done 106

10. L104: No need to use abbreviation such as UBOS, ICFPNFPs, PFPs, TCMPs as these abbreviations are not used after this. Advice has been taken; all these abbreviations have been deleted 133 - 134, 106 - 109

11. L119: Instead of “Generally,” you may write “In Uganda,” Thank you, this has been done 125

12. L136: …(UDHS)? Any way to rewrite it? We have rewritten it to “This was a secondary data analysis of nationally representative cross-sectional survey data conducted between June and December 2016.” 142 -143

13. L136 …secondary analysis or secondary data analysis? We have revised it to secondary data analysis 142

14. L139: In our secondary data analysis, we used data that were obtained from validated women’s… It is better to use “we” instead of using passive tense in the method section. Please recheck your whole method section from L139 to L189. Thank you, we have revised this section accordingly 148 - 206

15. L152: Again, “continuum of maternal care” or “continuum of care”? We have chosen to use “Continuum of maternity care” throughout the document. Whole document

16. L163: …exposure to newspapers/magazines,… We have revised this to media (Newspapers, media….. 169 - 170

17. L179: …proportions…Use of capital letters is not appropriate. Sorry about this, it has been corrected appropriately 185

18. L196, 197, 198: 95% CI can be cut as it is available from the Table1. These have been deleted 201 - 205

19. L202: Table 1: Percent (%) and 95% CI?

 Sorry for the omission, this has been corrected 208

20. L207-L215: …“continuum of maternal care” or “continuum of care”? We have chosen to use “Continuum of maternity care” throughout the document. Whole document

21. L226: …“continuum of maternal care” or “continuum of care”? We have chosen to use “Continuum of maternity care” throughout the document. Whole document

22. L227: In this context, 10.7% is better. 11% has been changed to 107% 233

23. L225-237: The coverage of the continuum of care in different countries followed the same definition? Are they comparable? All of them were measuring “continuum of maternal care” or “continuum of care”? Yes, they are comparable because they used the same definition for the continuum of maternity care. We crosschecked this in response to a comment by the reviewer from previous round of peer review comments. 234 - 243

24. L232: 10.0% in Tanzania? “level of coverage” has been added to make it complete. 238

25. L233: 8.0% in Ghana? “level of coverage” has been added to make it complete. 240

26. L234: 7.0% in Lao PDR? “level of coverage” has been added to make it complete. 240

27. L234: Lao should be Lao PDR. This change has been implemented 240

28. L237: 36.0%? 36.0% is the level of coverage in Western Kenya 243-244

29. L250 to 255: This paragraph is well written. It is optional, but it might be better to write a recommendation not in each paragraph but at the end. The current one is somehow acceptable, though. Thank you for the comment, we have chosen to leave it here since it is acceptable. 259 - 264

30. L264-268: This sentence is redundant and it is better not to use a word ‘fact’ in this context. This sentence “This low level of service utilisation could also be related to fact that many health facilities in Uganda do not have a clearly designated space for postnatal care as is the case with the ANC and childbirth.” Has been deleted. 275 - 277

31. L275: A new phrase appears in here, “continued care.” Please use the same expression. We have chosen to use “Continuum of maternity care” throughout the document. 284 -285

32. L307-309: This topic sentence does not seem to make sense. Please discuss your data, instead of discussing ref 8, 38 and 40. Thank you for advice, this has been revised accordingly 320 - 322

33. L312: …“don’t” should be “do not”. This has been changed 321

34. L316: …radio or TV? Or both? In the results, it was TV only. Thank you for the observation, it should be radio …… 325

35. L312: Are you discussing maternal and newborn health or only maternal health? As earlier suggested, we have adjusted it to maternal health 324

36. L329: What do you mean by print? Newspapers and magazines? Or either one? This has been clarified, it means both Newspapers and magazines 338

37. L329: …television or TV? We have replaced TV with television 170,213 & 221

38. L329: …“continuum of maternal care” or “continuum of care” or “continuum of care for maternal and newborn health”? We have chosen to use “Continuum of maternity care” throughout the document. 341

39. L331: secondary analysis or secondary data analysis? The nuance is different. We have revised it to analysis of secondary data 340

40. L335: You mean study limitations? This should be a separate subheading like “Conclusion.” Methodological considerations has been replace with “Study strengths and limitations” 344

41. L347-8: … “continuum of maternal care” or “continuum of care” or “continuum of care for maternal and newborn health”? We have chosen to use “Continuum of maternity care” throughout the document. 357 - 358

42. L351: “being a resident of western region was positively…” This is misleading. Thank you , the word “positively” has been removed 360

43. L352: …”ministry of health” or “Ministry of Health”? It should be “Ministry of Health” 364 -365

44. L352: Low levels of what kind of education? Why only northern and eastern regions? This has been revised to low level of formal education 363, 368 -369

45. L355-356: This is not a complete English sentence. We have now revised this sentence. 363 - 366

46. L356-357: Is this relevant as a conclusion of this article? This has been revised accordingly 367 - 369

47. Reference section should be checked one by one very carefully. For example, the journal name of the reference 3 is wrong. When you use PDF files as references, please add access dates. Please do not rely on computer soft wares for making the reference section. Advice taken and it has been implemented 383 -565

Reviewer #1:

Thank you for responding to my suggestions, and revising the manuscript. I saw the manuscript has been improved. However, I would to like to offer further suggestions to the authors Thank for the positive comments and advise NA

The descriptive result of place of childbirth, postnatal care, ANC visits, and complete continuum of care are the main finding of the study. I suggest the authors to make another table of these results to highlight them. Additionally, please show the results in the table with the following order: ANC visits, place of childbirth, postnatal care, and complete CoC. I found the data on ANC visits 4 and above is missed in Table 1. It is interesting to see the percentage of postnatal care among women delivered at facility and home, respectively, because the place of birth would affect the attendance of postnatal care.

I saw that the authors preferred to presenting percentage by column in Table. I, which I accepted. If presenting the percentage by column, it will be better to show the percentage of each category for women who did not complete CoC too, because readers are interested in difference in the distributions between women who completed and did not complete CoC. Thank you for the suggestions, it has been worked on and table 2 has been introduced to highlight the level of utilization of the different components of the COC.

The percentage of postnatal care among women delivered at facility and home, respectively has been added in the results section as advised.

The percentage of each category for women who did not complete CoC has also been added in Table 3. 223 - 224

Lines 216 - 219

246-247

Thank you for adding supplementary table 1. This is informative. Please briefly explain the calculations in the methods, results, and discussion sections. What do 56.3, 4.7, 44.2, and 10.9 mean in the cells of PNC 2001 and 2006. Why some results are reported with 95% CIs, and some are not. Please check missing values and clean the table. We thank you for this advice, it has been adopted in the current manuscript. The supplementary table has been revised as follows; - We have added the missing 95% CI of the major outcome variables.

- The confusing 56.3, 4.7, 44.2 and 10.9 for postnatal care have been removed

As suggested the same information has been included in the methods results and discussion sections. See supplementary table 1

See supplementary table 1

170 - 172

Line23, Line105: Regarding sampling methods, the authors mentioned multistage stratified sampling in the abstract and stratified two-stage cluster sampling in the methods section. Please use one. This has been corrected in the abstract 24

Line35, Line329: Please change print to newspaper. This has been changed to Newspaper and magazine 342

Line81: Delete a double quotation. The quotation has been removed 93

Line111: Please spell out 7.3M. This has been done 116

Line115-116: Please use a reference to explain improved access to emergency obstetric and neonatal care services.

 The reference has been provided 122

Line253: Please revise “every parturient should be …” to “every parturient and newborn should be…”

 Thank you for the suggestion, this has been done 262

Line278: The effect of early initiation of ANC is widely known. I suggest the authors to discuss more about why women do not attend the first ANC in the first trimester, and potential interventions to improve the situation. We done as suggested 317 - 319

Reviewer #2:

Thank you for addressing the reviewer’s comments. The revised paper is much improved, and the authors' responsiveness is much appreciated. I have only a few more comments: Thank for the positive comments and advise NA

1. Please mention when the MoH Uganda started recommending at least eight ANC visit in the introduction (lines 64-67). Also, explain the guideline on ANC visits when the DHS data were collected, which recommended at least four ANC visit, in either study settings or outcome variables. Thank you for the suggestions, these have been added appropriately as suggested. 76

165 - 167

2. The current manuscript requires a detailed review of formatting. Please spell out abbreviations only when they are used for the first time (e.g., ANC, PNC) This has been done throughout the document. NA

3. In “characteristics of the study population” in the results, I think you do not need to report exact numbers every time if you report percentages with their denominators, considering the word count. Please consider deleting exact numbers (i.e., 7,780/10,152). Thank for the advice, the repeated denominators have been deleted

 199 - 207

4. The “Study setting” seems a bit too lengthy (lines 101-134). The authors are encouraged to revise the subsection focusing on key information related to the continuum of care for maternal and newborn health in Uganda. Thank you for the suggestion, we revised it to remove some of the information. The extra description of the health care system in Uganda was requested for in the previous round of peer review. 117- 121

Reviewer #3: 

The authors modified the manuscript by addressing errors and added explanations and interpretations. However, please note that the mark-up copy of the revised manuscript attached to the submission file was not fully consistent with the clean version, which made difficult for review the revisions. In the Response-to-reviewers file, the line numbers for some point-to-point responses may not be correct. Sorry, about this confusion. We have revised this and hope that it is now resolved in the current document. NA

1. In the current submission, the authors presented the prevalence of individual components of services, namely ANC, place of delivery, and PNC, in Supplementary Table 1. Since the authors used these results extensively in the Discussion section, the authors seem to consider these results as important. The authors may want to describe the results of this supplementary table in the Results section.

 Thank you for the suggestion, this has been added 220 - 222

2. The authors seem to report the multivariable logistic regression analysis results of this study using Model II (N=8044) excluding those without currently married, not model I that included every eligible participant (N=10152). While the two models reported largely consistent results, the authors may want to use model I as the main result as it did not have loss of participants. They may use model II results as supplements (it simply shows that husbands’ education was not statistically significant and the results of multivariable regression analysis were not affected substantially after including husbands’ education and excluding participants without currently married. This has been worked on and Table 3 has been revised to remove adjusted model 11 which model 11 has been attached as a supplementary file 2 introduced in the methods.

The abstract, results sections and discussion have been revised to focus mainly on model 1 as the main results. Lines 36 – 47 and 205 – 206.

3. (minor reporting issue) The authors stated that “Our results show an increase of 16 percentage points between the attendance of four or more ANC visits and health facility delivery.” in the Discussion section (Lines 256-). The revised manuscript has Supplementary Table 1, which reported the figures at the decimal point. The coverage of ANC four times or more was 59.9% and that of health facility delivery was 76.6%. The difference may be 16.7 percentage points. The sentence above may be “… an increase of 17 percentage points….” Please check it. Thank you for pointing this out, we have revised it to 17% 265

END

---

## [Decision Letter · Decision Letter 2]

20 Oct 2021

PONE-D-21-14873R2Continuum of maternity care for maternal health in Uganda: a national cross-sectional study.PLOS ONE

Dear Dr. Milton W. Musaba,

Thank you for submitting your manuscript to PLOS ONE. After careful consideration, we feel that it has merit but does not fully meet PLOS ONE’s publication criteria as it currently stands. Therefore, we invite you to submit a revised version of the manuscript that addresses the points raised during the review process.

We look forward to receiving your revised manuscript.

Kind regards,

Masamine Jimba

Academic Editor

PLOS ONE

Journal Requirements:

Additional Editor Comments (if provided):

Thank you for your revision. I have checked your submission documents and found it very difficult to review it again. You are supposed to submit a set of clean and marked-up of the revised manuscript. However, 6 manuscripts are submitted and it is difficult to know which one to be reviewed. Please read the guideline and resubmit it again.

Reviewers' comments:

Reviewer's Responses to Questions

**Comments to the Author**

1. If the authors have adequately addressed your comments raised in a previous round of review and you feel that this manuscript is now acceptable for publication, you may indicate that here to bypass the “Comments to the Author” section, enter your conflict of interest statement in the “Confidential to Editor” section, and submit your "Accept" recommendation.

Reviewer #1: All comments have been addressed

Reviewer #2: All comments have been addressed

Reviewer #3: All comments have been addressed

2. Is the manuscript technically sound, and do the data support the conclusions?

Reviewer #1: Yes

Reviewer #2: Yes

Reviewer #3: Yes

3. Has the statistical analysis been performed appropriately and rigorously? 

Reviewer #1: No

Reviewer #2: Yes

Reviewer #3: Yes

4. Have the authors made all data underlying the findings in their manuscript fully available?

Reviewer #1: Yes

Reviewer #2: Yes

Reviewer #3: Yes

5. Is the manuscript presented in an intelligible fashion and written in standard English?

Reviewer #1: No

Reviewer #2: Yes

Reviewer #3: Yes

6. Review Comments to the Author

Reviewer #1: Thank you for your continuous effort to revise this manuscript. I confirmed that the authors have addressed my previous comments. However, I would like to suggest some more revisions.

According to the information on Revising Your Manuscript, a set of clean and marked-up copy of the revised manuscript should be submitted. However, a total of 6 manuscripts (3 were clean copies, and another 3 were marked-up copies) were compiled in the PDF for reviewer, which made me difficult to identify the one I should read this time.

Page 3. Please revise “22% seven million three hundred thousand (7.3 M)” to “22% (7.3 million)”

Page 4. Please delete a comma in the phrase of “A recent WHO report, estimates that”

Page 5. Independent variables: I think current use of modern contraception should be removed from the regression model, because “current” use of contraception cannot predict the study outcome that happened in the “past”. Thus, the current use of modern contraception is not an appropriate independent variable in this model. (Sorry for pointing out at this time. I should have recognized and suggested earlier.)

Page 6. Please delete the following parts because it is already mentioned: “ (four or more ANC contacts, health facility utilization during childbirth and postnatal care attendance)”

Page 6. Please report the percentage of complete continuum of maternity care for 2001, 2006, and 2011, respectively.

Table 1, 2, and 3. Please use the same font type and size across the tables.

Table1. Please delete 95%CI in the heading if actual data is not reported.

Table2. Please report all 95%CIs, or delete.

Page 11. Regarding “Given the limitations associated with demographic health survey data, there is need for a longitudinal study to follow a cohort of women from pregnancy through childbirth to the end of the postnatal period,” I think a longitudinal study is not necessarily needed to investigate the reason behind the sharp drop of service utilization between childbirth and postnatal care.

Reviewer #2: Dear authors

Thank you for addressing the reviewer’s comments. The revised paper is much improved. Please find a few more minor comments below:

1. Please report percentage only and delete the numerator and denominator indicated after the percentage in the first paragraph of the result section. For instance, please revise lines 219-221 “Among women who delivered at home, 29.4% (698/2382 95% CI: 28.0-31.7) had newborn PNC compared to 20.3% (1582/7780 95% CI: 19.2-21.0)”

2. Please review the first sentence in the conclusion: “In Uganda, less than one in ten (11%) women were able to utilise the entire continuum of maternity care for maternal and newborn health.” (lines 385-386). 11% is more than one in ten.

Reviewer #3: Thank you very much for address the reviewer's comments made in the previous round. The authors addressed most of the comments. The point that the authors and the reviewer may not agree upon is the treatment of supply-side factors by a more detailed region variable. I understand that previous articles controlled for a geographic factor by a broad category of regional variable (East; North; West; and Central). Since the authors employed analyses that accounted for a complex survey design of DHS, the remaining issue is how detail the geographic unit (as a fixed-effect variable) would be in the analysis. While I think more detailed geographic variables could estimate the coefficients for other independent variables with narrower variance (and confidence interval) in addition to capturing more detailed geographic differences in CoC, I respect the authors' decision.

7. PLOS authors have the option to publish the peer review history of their article (what does this mean?). If published, this will include your full peer review and any attached files.

Reviewer #1: No

Reviewer #2: No

Reviewer #3: **Yes: **Akira Shibanuma

---

## [Author Response · Author response to Decision Letter 2]

25 Oct 2021

25th /09/2021

To 

Dr. Masamine Jimba

Academic Editor

PLOS ONE

Dear Dr. Masamine Jimba

Re; Response to reviewers’ comments and resubmission of revised manuscript PONE-D-21-14873R2

Thank you for taking off time to review and provide feedback on this manuscript titled “Continuum of maternity care for maternal health in Uganda: a national cross-sectional study”. Please receive the revised manuscript and a point-by-point response to the comments raised by the peer reviewers as summarized in the table below. 

Comment Response Line

Additional Editor Comments

Thank you for your revision. I have checked your submission documents and found it very difficult to review it again. You are supposed to submit a set of clean and marked-up of the revised manuscript. However, 6 manuscripts are submitted and it is difficult to know which one to be reviewed. Please read the guideline and resubmit it again. Sorry, about this difficulty. Next time I will delete all the previous versions from the system and only leave the revised ones. Otherwise, we appreciate your commitment to improve this manuscript. NA

Reviewer #1:

Thank you for your continuous effort to revise this manuscript. I confirmed that the authors have addressed my previous comments. However, I would like to suggest some more revisions. We thank you for the positive comments and appreciation of our efforts to improve this manuscript. NA

According to the information on Revising Your Manuscript, a set of clean and marked-up copy of the revised manuscript should be submitted. However, a total of 6 manuscripts (3 were clean copies, and another 3 were marked-up copies) were compiled in the PDF for reviewer, which made me difficult to identify the one I should read this time. We are sorry about this mix up. It will be better next time. NA

Page 3. Please revise “22% seven million three hundred thousand (7.3 M)” to “22% (7.3 million)” Thank for the suggestion, this has been revised Line 96, page 3

Page 4. Please delete a comma in the phrase of “A recent WHO report, estimates that” This has been deleted Line 115, page 4

Page 5. Independent variables: I think current use of modern contraception should be removed from the regression model, because “current” use of contraception cannot predict the study outcome that happened in the “past”. Thus, the current use of modern contraception is not an appropriate independent variable in this model. (Sorry for pointing out at this time. I should have recognized and suggested earlier.) Thank you for this very important observation and we agree with you. It has been removed from the regression model. 

We have also revised table 3 and the rest of the manuscript Line 158, page 5

Line 225 – 226 ,page 9 – 10 

Page 6. Please delete the following parts because it is already mentioned: “ (four or more ANC contacts, health facility utilization during childbirth and postnatal care attendance)” Noted, this has been deleted Line 187 – 188, page 6

Page 6. Please report the percentage of complete continuum of maternity care for 2001, 2006, and 2011, respectively. Thank you for the suggestion, but we chose not to report zeros for 2001 and 2006 UDHS, while for 2011 and 2016 it is implied and specifying it would be repetition. We think that referring readers to the supplementary table 1 is sufficient. NA

Table 1, 2, and 3. Please use the same font type and size across the tables. This has been corrected Line 195 – 225 

Pages 6-10

Table1. Please delete 95%CI in the heading if actual data is not reported. We have deleted it Line 195, page 6

Table 2. Please report all 95%CIs, or delete. We have chosen to include all the 95% CI in this table Lines 198 – 199, page 8 

Page 11. Regarding “Given the limitations associated with demographic health survey data, there is need for a longitudinal study to follow a cohort of women from pregnancy through childbirth to the end of the postnatal period,” I think a longitudinal study is not necessarily needed to investigate the reason behind the sharp drop of service utilization between childbirth and postnatal care. Thank for this observation. Our suggestion of longitudinal cohort is in light of the fact that in the UDHS, participants interviewed may have used different components of the continuum of maternity care. For instance, a participant that never attended ANC and delivered at home may choose to attend postnatal care just to have her baby immunized. Lines 262 -263, page 13

Reviewer #2:

Dear authors, thank you for addressing the reviewer’s comments. The revised paper is much improved. Please find a few more minor comments below: We thank you for the positive comments and appreciation of our efforts to improve this manuscript NA

1. Please report percentage only and delete the numerator and denominator indicated after the percentage in the first paragraph of the result section. For instance, please revise lines 219-221 “Among women who delivered at home, 29.4% (698/2382 95% CI: 28.0-31.7) had newborn PNC compared to 20.3% (1582/7780 95% CI: 19.2-21.0)” We have deleted the extra information as suggested Line 185 – 190, pages 5 – 6.

2. Please review the first sentence in the conclusion: “In Uganda, less than one in ten (11%) women were able to utilise the entire continuum of maternity care for maternal and newborn health.” (lines 385-386). 11% is more than one in ten Thank you for the observation, we agree. This has been revised to one in ten Line 353, page 13

Reviewer #3

Thank you very much for address the reviewer's comments made in the previous round. The authors addressed most of the comments. The point that the authors and the reviewer may not agree upon is the treatment of supply-side factors by a more detailed region variable. I understand that previous articles controlled for a geographic factor by a broad category of regional variable (East; North; West; and Central). Since the authors employed analyses that accounted for a complex survey design of DHS, the remaining issue is how detail the geographic unit (as a fixed-effect variable) would be in the analysis. While I think more detailed geographic variables could estimate the coefficients for other independent variables with narrower variance (and confidence interval) in addition to capturing more detailed geographic differences in CoC, I respect the authors' decision. Thank you very much for your considered opinion. NA

END

---

## [Decision Letter · Decision Letter 3]

29 Nov 2021

PONE-D-21-14873R3Continuum of maternity care for maternal health in Uganda: a national cross-sectional study.PLOS ONE

Dear Dr. Milton W. Musaba,

Thank you for submitting your manuscript to PLOS ONE. After careful consideration, we feel that it has merit but does not fully meet PLOS ONE’s publication criteria as it currently stands. Therefore, we invite you to submit a revised version of the manuscript that addresses the points raised during the review process.

After several revisions, the quality is getting better, but it still needs further revisions. Please respond to the following comments.

The abstract of the cover page is an old one. Prevalence is still used, though it is corrected in the main text document.L50: I in 5,400…L59: …maternity care… In L61, 62: …maternity healthcare… These two are the same or different in nuance?L75: …maternal child health…  L78: maternal and child health (MCH)…These two are the same or different?L83: continuum of care for maternal health, L84: continuum of care, L86, 87: continuum of maternity care…The key words in this study is not appropriately used.L113: No need to abbreviate PNFPs.L119: This sentence is based on a WHO report, but the reference is very old (9 years ago), and not the WHO report.L123: ...secondary data analysis…L126, L131: …secondary analysis…L184…residents…L188: The details are shown in…L189: 29.4%)  ?L189: …table 1   L190: …Table 2   table of Table?L197, L221: Tables 2, 3. The use of capital letters are not appropriate.L224: Discussion is still redundant and confusing. The contents are getting better, but it is not easy to follow your discussion as paragraph wiring is very weak. Please try to include only one topic for each paragraph and do not discuss many different issues in each paragraph. Please make a summary of your main findings in the first paragraph of discussion section and do not discuss them. Then from the second paragraph, show your unique finding first in the first sentence of each paragraph as much as possible, and discuss your finding using your data and references. The flow should be specific to general.L292.293, 299: Eastern region… Western region? Central region?L343: Both maternal and newborn health? Previously, newborn health was not included. Why it suddenly appears here?L373: Reference is still not well done. For example, for ref 3, Lancet Glob Heal. Please check this section again. Please submit your revised manuscript by the end of this year. If you will need more time than this to complete your revisions, please reply to this message or contact the journal office at plosone@plos.org. Please include the following items when submitting your revised manuscript:A rebuttal letter that responds to each point raised by the academic editor and reviewer(s). You should upload this letter as a separate file labeled 'Response to Reviewers'.A marked-up copy of your manuscript that highlights changes made to the original version. You should upload this as a separate file labeled 'Revised Manuscript with Track Changes'.An unmarked version of your revised paper without tracked changes. You should upload this as a separate file labeled 'Manuscript'.

We look forward to receiving your revised manuscript.

Kind regards,

Masamine Jimba

Academic Editor

PLOS ONE

Journal Requirements:

Reviewers' comments:

Reviewer's Responses to Questions

**Comments to the Author**

1. If the authors have adequately addressed your comments raised in a previous round of review and you feel that this manuscript is now acceptable for publication, you may indicate that here to bypass the “Comments to the Author” section, enter your conflict of interest statement in the “Confidential to Editor” section, and submit your "Accept" recommendation.

Reviewer #1: All comments have been addressed

Reviewer #2: All comments have been addressed

2. Is the manuscript technically sound, and do the data support the conclusions?

Reviewer #1: Yes

Reviewer #2: Yes

3. Has the statistical analysis been performed appropriately and rigorously? 

Reviewer #1: Yes

Reviewer #2: Yes

4. Have the authors made all data underlying the findings in their manuscript fully available?

Reviewer #1: (No Response)

Reviewer #2: Yes

5. Is the manuscript presented in an intelligible fashion and written in standard English?

Reviewer #1: Yes

Reviewer #2: Yes

6. Review Comments to the Author

Reviewer #1: Thank you for addressing my previous comments. I saw the revised manuscript is much improved.

Please find a few more minor comments.

Table1. I am wondering if the authors might have forgot to delete the descriptive statistic of contraceptive use.

Table2. Please add a heading for frequency, percentage, and 95% confidence interval so that the term of 95% confidence interval in each cell can be deleted.

Tabl3. Please add a heading for frequency and percentage or (n, %) at the top of the columns of Incomplete CoC and Complete COC.

L274 in the clean file: Please delete current use of modern contraceptives.

Reviewer #2: I thank the authors for addressing my comments. They addressed all the comments, and the manuscript improved.

7. PLOS authors have the option to publish the peer review history of their article (what does this mean?). If published, this will include your full peer review and any attached files.

Reviewer #1: No

Reviewer #2: No

---

## [Author Response · Author response to Decision Letter 3]

21 Dec 2021

21st /12/2021

To 

Dr. Masamine Jimba

Academic Editor

PLOS ONE

Dear Dr. Masamine Jimba

Re; Response to reviewers’ comments and resubmission of revised manuscript PONE-D-21-14873R2

Thank you for taking off time to review and provide feedback on this manuscript titled “Continuum of maternity care for maternal health in Uganda: a national cross-sectional study”. Please receive the revised manuscript and a point-by-point response to the comments raised by the peer reviewers as summarized in the table below. 

Comment Response Line

Additional Editor Comments

After several revisions, the quality is getting better, but it still needs further revisions. Please respond to the following comments. We appreciate all your efforts to improve this manuscript. NA

The abstract of the cover page is an old one. Prevalence is still used, though it is corrected in the main text document. We have cross checked this and the abstract is up to date. Prevalence appears only twice in the whole document and not in the abstract. 14 – 43 

L50: I in 5,400… This has been fully spelt out 50

L59: …maternity care… In L61, 62: …maternity healthcare… These two are the same or different in nuance? We have chosen to stick to maternity care 61- 62 

L75: …maternal child health… L78: maternal and child health (MCH)…These two are the same or different? Noted, we have chosen to use maternal child health 78

L83: continuum of care for maternal health, L84: continuum of care, L86, 87: continuum of maternity care…The key words in this study is not appropriately used. Sorry for this mix up, we have revised this to the study key words “continuum of maternity care” 83 – 87 

L113: No need to abbreviate PNFPs. This abbreviation has been deleted 

L119: This sentence is based on a WHO report, but the reference is very old (9 years ago), and not the WHO report. Thank you for the observation, it has been corrected 469

L123: ...secondary data analysis…L126, L131: …secondary analysis… We have revised this to make it uniform “secondary data analysis” 123 – 131 

L184…residents… It has been corrected to “residents” 184

L188: The details are shown in… Shown has been included 189 – 190 

L189: 29.4%)? This was not put in the table. In one of the previous reviews, one review asked for this and we added it in the text. PNC for home deliveries was 698/2372, 29.4 (28.0-31.7) while no PNC was 1674/2372 hence 70.6 (68.3-72.0). PNC for health facility delivery was 1582/7780, 20.3 (19.2-21.0) and NO PNC was 6199/7780, 79.7 (79.0-80.8) NA

L189: …table 1 L190: …Table 2 table of Table? This is not clear to this, but it is probably related to the table above 209 – 210 

L197, L221: Tables 2, 3. The use of capital letters are not appropriate. This has been revised and capital letters removed from table 2 and 3. 233 – 235 

L224: Discussion is still redundant and confusing. The contents are getting better, but it is not easy to follow your discussion as paragraph wiring is very weak. Please try to include only one topic for each paragraph and do not discuss many different issues in each paragraph. Please make a summary of your main findings in the first paragraph of discussion section and do not discuss them. Then from the second paragraph, show your unique finding first in the first sentence of each paragraph as much as possible, and discuss your finding using your data and references. The flow should be specific to general. We have revised this section accordingly to improve the flow. 224 – 380 

L292.293, 299: Eastern region… Western region? Central region? The capital letters have been revised 292

L343: Both maternal and newborn health? Previously, newborn health was not included. Why it suddenly appears here? Sorry, “newborn” has been removed as requested earlier. 343

L373: Reference is still not well done. For example, for ref 3, Lancet Glob Heal. Please check this section again. We have revised this section 374 – 572 

Journal requirements

Please review your reference list to ensure that it is complete and correct. If you have cited papers that have been retracted, please include the rationale for doing so in the manuscript text, or remove these references and replace them with relevant current references. Any changes to the reference list should be mentioned in the rebuttal letter that accompanies your revised manuscript. If you need to cite a retracted article, indicate the article’s retracted status in the References list and also include a citation and full reference for the retraction notice. Thank you, we have revised and updated our reference list 374 – 572

Reviewer #1:

Thank you for addressing my previous comments. I saw the revised manuscript is much improved. Please find a few more minor comments. We appreciate all your efforts to improve this manuscript. NA

Table1. I am wondering if the authors might have forgot to delete the descriptive statistic of contraceptive use. Sorry about this oversight, we have deleted it 194 – 197 

Table2. Please add a heading for frequency, percentage, and 95% confidence interval so that the term of 95% confidence interval in each cell can be deleted. We have added this information 197 – 198 

Tabl3. Please add a heading for frequency and percentage or (n, %) at the top of the columns of Incomplete CoC and Complete COC. This has been added 222 – 223 

L274 in the clean file: Please delete current use of modern contraceptives. Sorry about this oversight, we have deleted it 274

Reviewer #2:

I thank the authors for addressing my comments. They addressed all the comments, and the manuscript improved. We appreciate all your efforts to improve this manuscript. NA

Reviewer #3

Overall comment; Overall, the authors discussed the important topic of the continuum of care for maternal and newborn health in Uganda. They conducted a secondary analysis of the 2016 Demographic and Health Survey to determine the prevalence and factors associated with the continuum of maternity care in Uganda. There was a general lack of interpretation of results regarding the discontinuities between antenatal care, health facility childbirth, and postnatal care services. The associated factors, such as initiation of ANC in the first trimester, level of education, and exposure to mass media, have been reported in several studies. Therefore, it would be great if the authors could add evidence and implication for policy makers based on findings of this study. Please find my comments in blue. -We appreciate all your efforts to improve this manuscript.

-Thank you for the observation regarding the discontinuities in care. We were concerned about the same finding, so we tried to explain and interpret the same in the discussion. 253 – 289 

Abstract 

Results: About 59.9% (6080/7262) (95% CI: 59.0-60.8) had four or more antenatal visits. -Please check if the denominator is correct. I think you do not need to report exact numbers every time if you report results with a percentage with the denominator. Noted the denominator should be 10,152.

We have chosen to delete the denominator, since it is uniform across the results reported. 28 – 31 

Introduction

The first paragraph seems too long. It needs to be restructured to provide an overview of maternal, neonatal and under-5 childhood deaths by comparing sub-Saharan Africa with the other regions of the world. We have revised this section as advised 45 – 59 

Almost all 277,300 (94%) of the annual maternal deaths occur in low and middle-income countries and roughly two-thirds (196,000) of these deaths happen in South East Asia and sub- Saharan Africa [2,3]. - How many annual maternal deaths occurred? When you report percentages, please indicate their denominators together. Thank you for this observation, we have revised it accordingly. 45 – 49 

It is estimated that 45% of all the under-5 childhood deaths happen early in the neonatal period, especially in the first seven (7) days of life. - Is it the global situation or and sub-Saharan Africa? These are global deaths 55

In introduction, you need to describe the situations of Uganda in terms of maternal, neonatal and under-5 childhood deaths if available. It is also recommended to include country data related to the study objectives briefly, such as the number of women of reproductive age, ANC and PNC attendance. It would provide rationales for the study objectives and the interventions for the continuum of maternity care recommended in the discussion. Thank you for the advice, we have added this information to the introduction.

-Country data related to study objectives was provided under the section on study setting 54 – 57 

96 – 125 

The current study focused on the completion of the continuum of care by specifically looking at antenatal care (ANC) attendance, health facility utilization during childbirth, and postnatal care (PNC). – This sentence in the second paragraph should be moved to the next paragraph as it describes the study objectives. We feel that this sentence is in the right place as per your suggestion. 88 – 90 

It is better to include evidence available on the association between the continuum of maternity care and maternal and newborn health if available in the second paragraph. Thank you, this is provided in the introduction 61 – 62 

84 – 87 

Materials and Methods

Study design: it should be mentioned that this study is the secondary analysis of the nationally-representative cross-sectional study (UDHS) conducted between June and December 2016. We have indicated this, thank you 123 – 126 

Sample size and sampling: Did all the sample of 10,152 women have the outcome data? If you excluded some women due to missing data on the outcome, please explain it or add a study population flow chart. We only had one variable that had missing data fit for multivariable analysis and how missing data was handled was explained in the statistical analysis.

Furthermore, the missing data was also indicated at the end of table 1 178 – 182 

Outcome Variable: Complete continuum of care was the outcome variable. Complete continuum of maternity care was considered when a woman reported having had: at least four ANC (ANC4+) contacts during the most recent pregnancy, utilized a health facility during the most recent childbirth and had at least one postnatal check-up within six weeks after childbirth: It would be great if you could add definitions of each segment consisting continuum of care. Are they in line with the Uganda MOH/WHO’s recommendations, for instance, in terms of the number and quality of ANC visits, supervision of a skilled service provider and length of stay in a health facility during childbirth and timing of postnatal check-up? Thank you for the comment and observation. We aimed at utilization of the available services, and not the quality of care, so we did not analyze content/quality/timing of ANC, PNC. This information is provided under the outcome variables section.

 140 -149

Independent variables: Some variables included in the analyses and the result tables were not listed here, such as contraceptive use and religion. For variable selection, it is better to refer to a conceptual framework, such as Owilli’s continuum of care framework for maternal, newborn and child health (https://doi.org/10.1186/s12889-016-3075-0). -We included all the variables in the methods section as they are in table1 and 3.

- Religion in the methods section has been highlighted. Regarding contraceptive use, we were advised to drop this from the study by one of the reviewers in the last review. 

- Use of Owilli’s continuum of care framework was advised in the earlier reviews and its indicated in the statistical analysis. 162 – 185 

Results

Characteristics of the study population: If you describe 95% CI in the narrative part, it is better to include them in Table 1 as well. Also, recheck Table 1 as total numbers exceed the sample size for some variables (i.e., postnatal care, residence). -We have cross checked our table 1, and all variables including residence and postnatal care in table 2 add up to 10,152. 

-We don’t think that it is necessary to include confidence intervals in table 1 because the table will become congested and it is not standard practice 198 – 200 

Of the 10,125 women, 59.9% (6,080/7262) (95% CI: 59.0-60.8) had four or more ANC contacts while 76.6% (7,780/10,125) (95% CI: 75.8-77.5) utilized health facility for childbirth and 22.5% (2,280/10,125) (95% CI: 21.5-23.2) attended postnatal care within six weeks after childbirth. – Is the denominator for four or more ANC visits correct? YES, it is, the denominator for all these percentages is 10,125. What is shown in the brackets are the numerators. 

 183 – 192 

Factors associated with complete continuum of maternity care: The description is too short. You need to explain the key statistical outputs provided in Table 2. Which model is the main model that you used to discuss the results? In the description, were those independent variables associated with the outcome negatively or positively? For this, I think it is important to include odds ratios for some key independent variables. Table 2 results were discussed under characteristics of the study population. This section of results focused on statistical outputs of table 3 and we included the main model in the statistical analysis section and also in the first statement of this section. We ensured to use more/less likely to have complete continuum of maternity care to show the positive/negative association. We included the odds ratios of all the significant variables under this section. 186 – 199 

202 – 219 

Throughout the manuscript, there were inconsistencies in reporting formats and typos, such as the use of thousand separators. Please review the whole manuscript. We have taken time to revise correct all these typos and other related errors. Quraish, please correct what you can see as well. NA

Discussion

We found a low level of utilization of complete continuum of maternity care at 11%. The level of use was highest for the place of childbirth at (7,789/10,125):77%), followed by ANC 4 or more visits at (6,080/7,262: 60%) and lowest for PNC at (2,280/1, 0125: 23%). - The discussion of the study seems mixed with results. This sentence was already presented in the results. You need to add more interpretation of the findings in the discussion. Thank you for the advice, we have done this in the revised manuscript 242 – 368 

Uganda’s levels of completion of the continuum of maternity care is similar to that of other countries in the region. For both Ethiopia and Tanzania, the reported level is at about 10% [27,29], 8% for Ghana [28] and 7% for Lao [12]. Compared to Kenya, it is more than three times below the reported level of completion of the continuum of maternity care at 36% [32]. - You compared the prevalence of a complete continuum of maternity care among several countries. Did these studies use the same definition of the continuum of maternity as one used for this study? Moreover, are they all national-level studies? Please note the level of continuum of care may vary even within a country. Therefore, if you want to compare the prevalence of continued care with other studies, they should be nationally representative data. Otherwise, please specify locations, for instance, districts and regions. Thank so much for this important observation.

Yes, we paid attention to these differences in the discussion as we made comparisons with other countries. 230 – 246 

The largest dropout of 54% occurred between the stages of childbirth and the postnatal period. - I think it would be useful if you could provide more insights on this point by comparing it with the situations of other counties. For instance, how about the situation in Kenya? Note, we have done as advised in the revised manuscript. 248 – 254 

In the health sector, the decentralization policy has greatly improved physical access because it is estimated that 72% of the population lives within five kilometers of a health facility [34].This could partly explain the observed increase in the number health facility deliveries because the basic emergency obstetric care services have been brought closer to the community. - This paragraph is confusing. This study used only UDHS 2016 data, so you were not able to explore changes over time. If you want to discuss the increase in the number of health facility deliveries, you need to present the baseline. You did not compare longitudinal data and present any data about health facility delivery in the past in this manuscript. Decentralization policy may have increased all the ANC, health facility deliveries, PNC. This policy was not discussed before, so you need to explain this. Thank you for this observation, it is correct.

However, in response to comments from a reviewer in the previous round we provided a supplementary file showing trends in the level of continuum of care over the years. We refer to it in the first paragraph of this section. 232 – 232 

Our results show an increase of 16% between the attendance of four or more ANC visits and health facility delivery… On the other hand, the observed drop in utilization between childbirth and postnatal care could be a pointer to the quality of service received during childbirth and labour. - They are directly related to the continued care. It would be useful if you could elaborate on why these discontinuities occurred, apart from the poor quality of maternity care in Uganda. Thank you, we have suggested possible explanations for these discontinuities 248 – 284 

Most of the factors associated with the continued care, such as initiation of ANC in the first trimester, level of education, and exposure to mass media, were reported and examined in previous studies. It is great if you could add evidence to them based on the findings of this study. At the same time, you described each associated factor using one paragraph, which is a bit wordy. It is great you can summarize discussions and put more focus on the unique findings of your study. -Thank you so much for this comment. As much as other countries have looked at factors associated with continuum of care, we believe we cannot assume that factors shown in other studies apply to the Ugandan context. We have not come across any study in Uganda that has analyzed DHS data to assess factors associated with continuum of care.

-Furthermore, we have discussed the dropout between ANC and facility delivery and facility delivery and PNC.

-The key finding in this study is about the discontinuities at the different stages of care, which we have discussed extensively and identified gaps that we could not answer using the UDHS data. 229 – 373 

You found large geographic variations in the continuum of maternity care. It is interesting if you can further investigate why the Eastern area, the poorest region with some of the poorest maternal and child health indicators, had a better continuum of care than the Western area. For instance, are there development partners supporting the continuum of care in this region? This tends to attract more development partners in the region to support the provision of social services such as maternity care. 327 – 328 

This finding may partly be explained by the unique cultural practices and beliefs about pregnancy and childbirth in this region – Please provide a few examples here. This is one the examples that we have provided; “For instance, anecdotal evidence from this region suggests that for the first seven days after childbirth, the baby should not be seen by strangers. So, babies born at home are kept indoors until the first week of life has elapsed. While those that give birth normally in health facility are in a rush to get back home early to full fill this cultural practice because the health facility does not offer adequate privacy.” 321 – 328 

I think it would be great if you could include recommendations beyond mass media and education programs to promote the complete continuum of care, such as support for early initiation of ANC and the integration of ANC and PNC programs based on your findings. These have been included in the abstract and conclusion sections 41 – 43 

421 – 423 

END

---

## [Decision Letter · Decision Letter 4]

10 Jan 2022

PONE-D-21-14873R4Continuum of maternity care for maternal health in Uganda: a national cross-sectional study.PLOS ONE

Dear Dr. Milton W. Musaba,

Thank you for submitting your manuscript to PLOS ONE. After careful consideration, we feel that it has merit but does not fully meet PLOS ONE’s publication criteria as it currently stands. Therefore, we invite you to submit a revised version of the manuscript that addresses the points raised during the review process.

Sorry for my delayed response. December and January have been extremely busy. I checked your response sheet and found your response sheet is written for the previous revision comments (R2). I could not find your  response for the R3 version. Please check our comments again carefully and respond to them. As new comments are given from one reviewer, you may respond to them, too.

Please submit your revised manuscript by Feb 24 2022 11:59PM If you will need more time than this to complete your revisions, please reply to this message or contact the journal office at plosone@plos.org. Please include the following items when submitting your revised manuscript:A rebuttal letter that responds to each point raised by the academic editor and reviewer(s). You should upload this letter as a separate file labeled 'Response to Reviewers'.A marked-up copy of your manuscript that highlights changes made to the original version. You should upload this as a separate file labeled 'Revised Manuscript with Track Changes'.An unmarked version of your revised paper without tracked changes. You should upload this as a separate file labeled 'Manuscript'.If applicable, we recommend that you deposit your laboratory protocols in protocols.io to enhance the reproducibility of your results. Protocols.io assigns your protocol its own identifier (DOI) so that it can be cited independently in the future. For instructions see: https://journals.plos.org/plosone/s/submission-guidelines#loc-laboratory-protocols. Additionally, PLOS ONE offers an option for publishing peer-reviewed Lab Protocol articles, which describe protocols hosted on protocols.io. Read more information on sharing protocols at https://plos.org/protocols?utm_medium=editorial-email&utm_source=authorletters&utm_campaign=protocols.

We look forward to receiving your revised manuscript.

Kind regards,

Masamine Jimba

Academic Editor

PLOS ONE

Journal Requirements:

Reviewers' comments:

Reviewer's Responses to Questions

**Comments to the Author**

1. If the authors have adequately addressed your comments raised in a previous round of review and you feel that this manuscript is now acceptable for publication, you may indicate that here to bypass the “Comments to the Author” section, enter your conflict of interest statement in the “Confidential to Editor” section, and submit your "Accept" recommendation.

Reviewer #1: All comments have been addressed

2. Is the manuscript technically sound, and do the data support the conclusions?

Reviewer #1: Yes

3. Has the statistical analysis been performed appropriately and rigorously? 

Reviewer #1: Yes

4. Have the authors made all data underlying the findings in their manuscript fully available?

Reviewer #1: Yes

5. Is the manuscript presented in an intelligible fashion and written in standard English?

Reviewer #1: No

6. Review Comments to the Author

Reviewer #1: Table2: Please correct the heading of “Percentage (95%” to “Percentage (95% CI)”, and delete “95%CI:” in the cell of each result under the heading. I have another suggestion on Table 2; because this is the main finding, I recommend the authors to replace the Table 2 with a figure, which will be more attractive.

L254- 256, L269-270: These two sentences seem repetition.

L279-280. The initiation of ANC visits in the first trimester and completed the continuum of maternity care may both represent good care seeking behavior of women. Please consider this discussion too.

L292-298. This portion is not much related to the association between early initiation of ANC visit and completed continuum of care that is focused in this paragraph. I suggest the authors to make another paragraph for it.

L328-329. Although exposure to radio, newspaper or magazine are both associated with completed continuum of care, the impact of newspaper and magazine on the study outcome may be different from that of radio, because the use of

newspaper and magazine requires money and literacy. Please discuss this point too.

L355-366. This sentence has a line break in the middle. Please correct.

7. PLOS authors have the option to publish the peer review history of their article (what does this mean?). If published, this will include your full peer review and any attached files.

Reviewer #1: No

---

## [Author Response · Author response to Decision Letter 4]

19 Jan 2022

19th /01/2022

To 

Dr. Masamine Jimba

Academic Editor

PLOS ONE

Dear Dr. Masamine Jimba

Re; Response to reviewers’ comments and resubmission of revised manuscript PONE-D-21-14873R4

Thank you for taking off time to review and provide feedback on this manuscript titled “Continuum of maternity care for maternal health in Uganda: a national cross-sectional study”. Please receive the revised manuscript and a point-by-point response to the comments raised by the peer reviewers as summarized in the table below. 

Comment Response Line

Additional Editor Comments

Sorry for my delayed response. December and January have been extremely busy. I checked your response sheet and found your response sheet is written for the previous revision comments (R2). I could not find your response for the R3 version. Please check our comments again carefully and respond to them. As new comments are given from one reviewer, you may respond to them, too. Sorry, about the error in naming this file R2 instead of R3.

I would like to confirm that it is the correct one, but we were surprised that the comments from the reviewer #3 were similar to those we had responded to in the last two successive rounds of review. NA

Journal requirements

Please review your reference list to ensure that it is complete and correct. If you have cited papers that have been retracted, please include the rationale for doing so in the manuscript text, or remove these references and replace them with relevant current references. Any changes to the reference list should be mentioned in the rebuttal letter that accompanies your revised manuscript. If you need to cite a retracted article, indicate the article’s retracted status in the References list and also include a citation and full reference for the retraction notice. Thank you, we have revised and updated our reference list 374 – 572

Reviewer #1:

Table2: Please correct the heading of “Percentage (95%” to “Percentage (95% CI)”, and delete “95%CI:” in the cell of each result under the heading. I have another suggestion on Table 2; because this is the main finding, I recommend the authors to replace the Table 2 with a figure, which will be more attractive. Thank you for the suggestion to replace table 2 with a figure. However, from the previous reviews, we were requested to have this table that we have improved overtime based on several recommendations of the reviewers. So, we kindly request to keep it. 207 – 208 

L254- 256, L269-270: These two sentences seem repetition. We think that they are related but not the same. The second sentence explains the positive association between GDP and socioeconomic status. 252 -257

L279-280. The initiation of ANC visits in the first trimester and completed the continuum of maternity care may both represent good care seeking behavior of women. Please consider this discussion too. We have elaborated further on this point 310 - 329

L292-298. This portion is not much related to the association between early initiation of ANC visit and completed continuum of care that is focused in this paragraph. I suggest the authors to make another paragraph for it. Thank you for the suggestion, we have created a new paragraph. 325 - 344

L328-329. Although exposure to radio, newspaper or magazine are both associated with completed continuum of care, the impact of newspaper and magazine on the study outcome may be different from that of radio, because the use of newspaper and magazine requires money and literacy. Please discuss this point too. We have elaborated further on this point 375-394

L355-366. This sentence has a line break in the middle. Please correct. We have revised this accordingly to make two separate paragraphs 351 – 360

361 – 380 

END

---

## [Decision Letter · Decision Letter 5]

26 Jan 2022

PONE-D-21-14873R5Continuum of maternity care for maternal health in Uganda: a national cross-sectional study.PLOS ONE

Dear Dr. Milton W. Musaba,

Thank you for submitting your manuscript to PLOS ONE. After careful consideration, we feel that it has merit but does not fully meet PLOS ONE’s publication criteria as it currently stands. Therefore, we invite you to submit a revised version of the manuscript that addresses the points raised during the review process.

It is getting better. Please do not give up and do you your best to improve the manuscript.

We look forward to receiving your revised manuscript.

Kind regards,

Masamine Jimba

Academic Editor

PLOS ONE

Additional Editor Comments:

Thank you for your revision for this time. The quality of manuscript is much better than before. Please make more efforts to improve it much better. Your basic writing skills have improved, but there is a room for further improvement.

1. L25: …multivariable logistic regression analysis to determine…

2. L37: Eastern Uganda, and exposure to …

3. L48: Ref [4]. Please check if this reference really correct. The reference title suggests that this is not a correct citation. Please check all the references, too, if they are correctly cited.

4. L67: ...at least eight antenatal care (ANC)…Please abbreviate it when it first appears. This is a basic rule for using abbreviation.

5. L79: …unsatisfactory maternal and child health (MCH)….

6. L80-81: …such as ANC, skilled….

7. L84: ….looking at ANC…Once it is abbreviated, no need to do it again.

8. L85: …childbirth and PNC. We aimed….

9. L92: The Uganda Demographic and Health Survey (UDHS)…

10. L123: …the secondary data analysis of…

11. L124 or L125?: Study population… Please check the text carefully.

12. L138-139: …maternity care. Its definition was based on the WHO….

13. L145: What is CoC?

14. L169: …conducted a bivariate logistic regression analysis…Please add “analysis” for other similar parts.

15. L177:…(AOR), 95% CI and p-values…

16. L187: …(2,280)… Please close a parenthesis appropriately.

17. L188: …(1,091) … Please close a parenthesis appropriately.

18. L200:…regression analysis…

19. L229-230: This sentence suggest that you have data about continuum of maternity care in 2011. So, this study is not the first time showing the level of the completion of the continuum of maternity care? Please clarify this point.

20. L229: Please show that the completion rate was about 11%.

21. L239: This could be partly because…This sentence does not show Ghana and Lao PDR are similar. They are also middle income countries.

22. L240: (GDP), while…

23. L242: World Bank…Please be more careful about the use of capital letters.

24. L245, 246: Is this sentence your finding? This paragraph is too long. Please show your finding first and do not over discuss it. You may cut many sentences in this paragraph.

25. L302, L313: “central region” or “Central Uganda”?

26. L303: “eastern region” or “Eastern Uganda”?

27. L303, L310:“western Uganda” or “Western Uganda”?

28. L305: …region of the country. Therefore…

29. L312: …the country and its MCH indicators…

30. L320: …the poorest MCH indicators…

31. L341: …for better MCH in the country. Sometimes MCH covers newborn health in some context.

32. L366: In Uganda, about 11% of women completed the continuum of maternity care,…

33. L368:…utilization of PNC services.

34. L373:…integration of all MCH services may go a long way in improving the continuum of maternity care.

35. L375-6:…towards improved MCH.

36. L412: Check the journal name again.

37. L415: Udhs or UDHS?

38. L497: Check the abbreviation.

39. L501. Check the abbreviation.

40. L508: Check the abbreviation.

41. L544: Check the abbreviation.

Reviewers' comments:

Reviewer's Responses to Questions

**Comments to the Author**

1. If the authors have adequately addressed your comments raised in a previous round of review and you feel that this manuscript is now acceptable for publication, you may indicate that here to bypass the “Comments to the Author” section, enter your conflict of interest statement in the “Confidential to Editor” section, and submit your "Accept" recommendation.

Reviewer #1: (No Response)

2. Is the manuscript technically sound, and do the data support the conclusions?

Reviewer #1: Yes

3. Has the statistical analysis been performed appropriately and rigorously? 

Reviewer #1: Yes

4. Have the authors made all data underlying the findings in their manuscript fully available?

Reviewer #1: Yes

5. Is the manuscript presented in an intelligible fashion and written in standard English?

Reviewer #1: Yes

6. Review Comments to the Author

Reviewer #1: Thank you for addressing my previous comments.

I have one last comment on Table 2.

Let me ask once again for deleting "95%CI" before 59.0, 39.2, 75.8, 22.5, 21.5, 76.8, 10.0, and 88.8.

Because you have indicated (95%CI) at the heading of the table, readers understand that (95.0-60.8) indicates 95%CI.

Please look at Table 3 in which (95%CI) is indicated at the heading of the table only, not for each result. This is what I requested for Table 2.

In this way, please keep in your mind that consistency across a manuscript is important for us researchers to deliver our findings appropriately.

7. PLOS authors have the option to publish the peer review history of their article (what does this mean?). If published, this will include your full peer review and any attached files.

Reviewer #1: No

---

## [Author Response · Author response to Decision Letter 5]

27 Jan 2022

27th /01/2022

To 

Dr. Masamine Jimba

Academic Editor

PLOS ONE

Dear Dr. Masamine Jimba

Re; Response to reviewers’ comments and resubmission of revised manuscript PONE-D-21-14873R5

Thank you for taking off time to review and provide feedback on this manuscript titled “Continuum of maternity care for maternal health in Uganda: a national cross-sectional study”. Please receive the revised manuscript and a point-by-point response to the comments raised by the peer reviewers as summarized in the table below. 

Comment Response Line

Additional Editor Comments

Thank you for your revision for this time. The quality of manuscript is much better than before. Please make more efforts to improve it much better. Your basic writing skills have improved, but there is a room for further improvement. Thank you for the encouragement and for your time and effort to make this manuscript better. We do not take it for granted. NA

1. L25: …multivariable logistic regression analysis to determine… We have added the missing word ‘analysis’ 25

2. L37: Eastern Uganda, and exposure to … We have added the missing word ‘and’ 37

3. L48: Ref [4]. Please check if this reference really correct. The reference title suggests that this is not a correct citation. Please check all the references, too, if they are correctly cited. This has been revised 48

388 – 616 

4. L67: ...at least eight antenatal care (ANC)…Please abbreviate it when it first appears. This is a basic rule for using abbreviation. Thank you for this observation, in L24 that is when ANC first appears in this manuscript and believe that it is correctly done. 68

5. L79: …unsatisfactory maternal and child health (MCH)…. We have added the missing word ‘and’ 79

6. L80-81: …such as ANC, skilled…. We have corrected this 81 – 82 

7. L84: ….looking at ANC…Once it is abbreviated, no need to do it again. We have corrected this 85

8. L85: …childbirth and PNC. We aimed…. We have corrected this 86

9. L92: The Uganda Demographic and Health Survey (UDHS)… We have corrected this 93

10. L123: …the secondary data analysis of… We have added the missing word ‘data’ 124 

11. L124 or L125?: Study population… Please check the text carefully. Sorry, this should be a separate subtitle. We have corrected it. 126

12. L138-139: …maternity care. Its definition was based on the WHO…. Thank you for the suggestion, we have revised it accordingly. 140 – 141 

13. L145: What is CoC? We have revised it now. 147 – 148 

14. L169: …conducted a bivariate logistic regression analysis…Please add “analysis” for other similar parts. We have added the missing word ‘analysis’ 25, 172, 203

15. L177:…(AOR), 95% CI and p-values… Corrected 180

16. L187: …(2,280)… Please close a parenthesis appropriately. We have closed it appropriately 190

17. L188: …(1,091) … Please close a parenthesis appropriately. We have closed it appropriately 191

18. L200:…regression analysis… We have added the missing word ‘analysis’ 203

19. L229-230: This sentence suggest that you have data about continuum of maternity care in 2011. So, this study is not the first time showing the level of the completion of the continuum of maternity care? Please clarify this point. We have presented that data for 2011 in the supplementary file. 234

20. L229: Please show that the completion rate was about 11%. We have revised this sentence to reflect this 232- 233 

21. L239: This could be partly because…This sentence does not show Ghana and Lao PDR are similar. They are also middle income countries. We have checked this again and these two countries are not yet middle-income countries.

https://opendevelopmentmekong.net/news/world-bank-classifies-laos-as-lower-middle-income-economy-for-2020/

https://mofep.gov.gh/press-release/2021-04-14/ghana-has-not-been-down-graded-as-a-low-income-country

NA

22. L240: (GDP), while…

 We have closed the parenthesis appropriately. 245

23. L242: World Bank…Please be more careful about the use of capital letters. Sorry about this, it has been corrected 246

24. L245, 246: Is this sentence your finding? This paragraph is too long. Please show your finding first and do not over discuss it. You may cut many sentences in this paragraph. Yes, this is our finding. We have revised this paragraph as suggested by cutting out several sentences. 249 - 293

25. L302, L313: “central region” or “Central Uganda”? Sorry about this, it has been corrected 315

26. L303: “eastern region” or “Eastern Uganda”? Sorry about this, it has been corrected 316

27. L303, L310:“western Uganda” or “Western Uganda”? Sorry about this, it has been corrected 316

28. L305: …region of the country. Therefore… Sorry about this, it has been corrected 318

29. L312: …the country and its MCH indicators… Corrected appropriately 326

30. L320: …the poorest MCH indicators… Corrected 334

31. L341: …for better MCH in the country. Sometimes MCH covers newborn health in some context. Thank you for the clarification, we have corrected this 354

32. L366: In Uganda, about 11% of women completed the continuum of maternity care,… Thank you for the suggestion, we have incorporated it 379

33. L368:…utilization of PNC services. We have corrected this. 381

34. L373:…integration of all MCH services may go a long way in improving the continuum of maternity care. We have corrected this. 387

35. L375-6:…towards improved MCH. We have corrected this. 389

36. L412: Check the journal name again. We have corrected the name of the journal, thank you for the observation 425

37. L415: Udhs or UDHS? This has been corrected now 427

38. L497: Check the abbreviation We have checked it 510

39. L501. Check the abbreviation. We have checked it 514

40. L508: Check the abbreviation. We have checked it 521

41. L544: Check the abbreviation. We have checked it 557

Reviewer #1:

Thank you for addressing my previous comments. I have one last comment on Table 2. We are very grateful for all the time and effort that you have dedicated to make this manuscript better. NA

Let me ask once again for deleting "95%CI" before 59.0, 39.2, 75.8, 22.5, 21.5, 76.8, 10.0, and 88.8.

Because you have indicated (95%CI) at the heading of the table, readers understand that (95.0-60.8) indicates 95%CI. Okay, sorry about this inaction. We totally agree with your suggestion. I think the track changes were too many in the previous version and we could not easily see this repetition 200 – 201 

Please look at Table 3 in which (95%CI) is indicated at the heading of the table only, not for each result. This is what I requested for Table 2.

In this way, please keep in your mind that consistency across a manuscript is important for us researchers to deliver our findings appropriately. Point taken, thank you very much for consistently pointing this out. 

END

---

## [Decision Letter · Decision Letter 6]

2 Feb 2022

PONE-D-21-14873R6Continuum of maternity care for maternal health in Uganda: a national cross-sectional study.PLOS ONE

Dear Dr. Milton W. Musaba,

Thank you for submitting your manuscript to PLOS ONE. After careful consideration, we feel that it has merit but does not fully meet PLOS ONE’s publication criteria as it currently stands. Therefore, we invite you to submit a revised version of the manuscript that addresses the points raised during the review process.

One more revision please. The goal is almost there.

We look forward to receiving your revised manuscript.

Kind regards,

Masamine Jimba

Academic Editor

PLOS ONE

Journal Requirements:

Additional Editor Comments:

Thank you for your revision. Though most of comments were well-addressed, it still needs further revisions: one major and several minor ones.

L240: Please read the World Bank home page (or a blog) carefully.

https://blogs.worldbank.org/opendata/new-world-bank-country-classifications-income-level-2020-2021

“The World Bank assigns the world’s economies to four income groups—low, lower-middle, upper-middle, and high-income countries.”

Middle income countries are divided into two：lower middle and upper-middle. So lower-middle income countries are aprt of middle income countries and it is different from low income counties.

You quoted two references in your response sheet.

a. “The World Bank has classified Laos as a lower-middle-income economy in its latest classification for the 2020 fiscal year.”

https://opendevelopmentmekong.net/news/worldbank-

classifies-laos-as-lower-middle-incomeeconomy-

for-2020/

b. Ghana has not been down-graded as a low income country.

https://mofep.gov.gh/press-release/2021-04-

14/ghana-has-not-been-down-graded-as-a-lowincome-

country

So both are middle income countries.

L356: …completed the continuum of maternity care. In academic literature, an expression “maternal care” is often used. Please check which is better “maternity care” or “maternal care,” throughout the manuscript.

L386: Lancet Glob Heal should be Lancet Glob Health.

L400: As other references are abbreviated, this journal name should be “correctly” abbreviated.

L485: World Heal Organ. It should be World Health Organization.

L496: BMJ Glob Heal should be BMJ Glob Health.

L552: BMJ Glob Heal should be BMJ Glob Health.

L577: Front Public Heal should be Front Public Health.

L586: Communicatio. Is it correct?

L604: Trop Med Int Heal should be Trop Med Int Health.

Please check references one by one with your own eyes without depending on computer soft wares.

Reviewers' comments:

Reviewer's Responses to Questions

**Comments to the Author**

1. If the authors have adequately addressed your comments raised in a previous round of review and you feel that this manuscript is now acceptable for publication, you may indicate that here to bypass the “Comments to the Author” section, enter your conflict of interest statement in the “Confidential to Editor” section, and submit your "Accept" recommendation.

Reviewer #1: All comments have been addressed

2. Is the manuscript technically sound, and do the data support the conclusions?

Reviewer #1: Yes

3. Has the statistical analysis been performed appropriately and rigorously? 

Reviewer #1: Yes

4. Have the authors made all data underlying the findings in their manuscript fully available?

Reviewer #1: Yes

5. Is the manuscript presented in an intelligible fashion and written in standard English?

Reviewer #1: Yes

6. Review Comments to the Author

Reviewer #1: I thank the authors for revising their manuscript six times. The authors have addressed all my suggestions. Congratulations on your achievement.

7. PLOS authors have the option to publish the peer review history of their article (what does this mean?). If published, this will include your full peer review and any attached files.

Reviewer #1: No

---

## [Author Response · Author response to Decision Letter 6]

4 Feb 2022

07th /02/2022

To 

Dr. Masamine Jimba

Academic Editor

PLOS ONE

Dear Dr. Masamine Jimba

Re; Response to reviewers’ comments and resubmission of revised manuscript PONE-D-21-14873R6

Thank you for taking off time to review and provide feedback on this manuscript titled “Continuum of maternity care for maternal health in Uganda: a national cross-sectional study”. Please receive the revised manuscript and a point-by-point response to the comments raised by the peer reviewers as summarized in the table below. 

Comment Response Line

Additional Editor Comments

Thank you for your revision. Though most of comments were well-addressed, it still needs further revisions: one major and several minor ones. Thank you for the encouragement and for your time and effort to make this manuscript better. We do not take it for granted. NA

L240: Please read the World Bank home page (or a blog) carefully.

https://blogs.worldbank.org/opendata/new-world-bank-country-classifications-income-level-2020-2021

“The World Bank assigns the world’s economies to four income groups—low, lower-middle, upper-middle, and high-income countries.”

Middle income countries are divided into two：lower middle and upper-middle. So lower-middle income countries are aprt of middle income countries and it is different from low income counties.

You quoted two references in your response sheet.

a. “The World Bank has classified Laos as a lower-middle-income economy in its latest classification for the 2020 fiscal year.”

https://opendevelopmentmekong.net/news/worldbank-

classifies-laos-as-lower-middle-incomeeconomy-

for-2020/

b. Ghana has not been down-graded as a low income country.

https://mofep.gov.gh/press-release/2021-04-

14/ghana-has-not-been-down-graded-as-a-lowincome-

country.

So both are middle income countries. Thank you for the clarification, this paragraph has been revised and we think that the anomaly has been corrected. 235 – 256 

L356: …completed the continuum of maternity care. In academic literature, an expression “maternal care” is often used. Please check which is better “maternity care” or “maternal care,” throughout the manuscript. We have revised this throughout the document NA

L386: Lancet Glob Heal should be Lancet Glob Health. Revised 399

L400: As other references are abbreviated, this journal name should be “correctly” abbreviated.

 Revised 413

L485: World Heal Organ. It should be World Health Organization. Revised 498

L496: BMJ Glob Heal should be BMJ Glob Health. Revised 509

L552: BMJ Glob Heal should be BMJ Glob Health.

 Revised 565

L577: Front Public Heal should be Front Public Health.

 Revised 590

L586: Communicatio. Is it correct? Revised 599

L604: Trop Med Int Heal should be Trop Med Int Health. Corrected 608

Please check references one by one with your own eyes without depending on computer soft wares. Thank you for the advise this has been done NA

END

---

## [Editor Report · Decision Letter 7]

7 Feb 2022

Continuum of care for maternal health in Uganda: a national cross-sectional study.

PONE-D-21-14873R7

Dear Dr. Milton W. Musaba,

We’re pleased to inform you that your manuscript has been judged scientifically suitable for publication and will be formally accepted for publication once it meets all outstanding technical requirements.

Kind regards,

Masamine Jimba

Academic Editor

PLOS ONE

Additional Editor Comments (optional):

Congratulations! I appreciate your hard work to revise this manuscript for seven times. It still needs minor revisions: in some parts, you are still using "maternity" instead of "maternal". You may use search engine to check it. However, you can do it at proof reading stage. I hope you can become more productive in academic research in the coming future.
---

## [Editor Report · Acceptance letter]

15 Feb 2022

PONE-D-21-14873R7 

Continuum of care for maternal health in Uganda: a national cross-sectional study. 

Dear Dr. Musaba:

I'm pleased to inform you that your manuscript has been deemed suitable for publication in PLOS ONE. Congratulations! Your manuscript is now with our production department. 

Kind regards, 

on behalf of

Professor Masamine Jimba 

Academic Editor

PLOS ONE